# Distinct release properties of glutamate/GABA co-transmission serve as a frequency-dependent filtering of supramammillary inputs

Himawari Hirai[1], Kohtarou Konno[2], Miwako Yamasaki[2], Masahiko Watanabe[2], Takeshi Sakaba[1], Yuki Hashimotodani[1]*

[1]Graduate School of Brain Science, Doshisha University, Kyoto, Japan; [2]Department of Anatomy, Faculty of Medicine, Hokkaido University, Sapporo, Japan

## eLife Assessment

This **fundamental** work provides evidence that glutamate and GABA are released from different synaptic vesicles at supramammillary axon terminals onto granule cells of the dentate gyrus. The study uses complementary electrophysiological and anatomical experimental approaches. Together, these provide **convincing** evidence that the co-release of glutamate and GABA from different vesicles within the same terminal could modulate granule cell firing in a frequency-dependent manner, although thorough elimination of alternative mechanisms would have strengthened the study. The work will be of interest to neuroscientists investigating co-release of neurotransmitters in various synapses in the brain and those interested in subcortical control of hippocampal function.

**\*For correspondence:**
hashimotodani@gmail.com

**Competing interest:** The authors declare that no competing interests exist.

**Abstract** Glutamate and GABA co-transmitting neurons exist in several brain regions; however, the mechanism by which these two neurotransmitters are co-released from the same synaptic terminals remains unclear. Here, we show that the supramammillary nucleus (SuM) to dentate granule cell synapses, which co-release glutamate and GABA, exhibit differences between glutamate and GABA release properties in paired-pulse ratio, $Ca^{2+}$-sensitivity, presynaptic receptor modulation, and $Ca^{2+}$ channel-vesicle coupling configuration. Moreover, uniquantal synaptic responses show independent glutamatergic and GABAergic responses. Morphological analysis reveals that most SuM terminals form distinct glutamatergic and GABAergic synapses in proximity, each characterized by GluN1 and $GABA_A\alpha1$ labeling, respectively. Notably, glutamate/GABA co-transmission exhibits distinct short-term plasticities, with frequency-dependent depression of glutamate and frequency-independent stable depression of GABA. Our findings suggest that glutamate and GABA are co-released from different synaptic vesicles within the SuM terminals, and reveal that distinct transmission modes of glutamate/GABA co-release serve as frequency-dependent filters of SuM inputs.

## Introduction

Despite the classical view of Dale's Principle known as 'one neuron, one transmitter' (*Eccles et al., 1954*), a growing body of evidence has demonstrated that many types of neurons in several brain regions co-release multiple neurotransmitters including classical neurotransmitters (glutamate, GABA, glycine, and acetylcholine), monoamines, purines, and neuropeptides (*Hnasko and Edwards, 2012*; *Vaaga et al., 2014*; *Trudeau and El Mestikawy, 2018*; *Wallace and Sabatini, 2023*). While this co-release is thought to contribute to several brain functions (*Nusbaum et al., 2001*; *Trudeau et al., 2014*;

*Tritsch et al., 2016*; *Ma et al., 2018*; *Kim and Sabatini, 2022*), little is known about the functional implications of the co-release of two or more neurotransmitters into neural circuits and the molecular mechanisms of how multiple neurotransmitters are co-transmitted. Here, we define co-release as the release of multiple neurotransmitters from a single neuron, regardless of whether the neurotransmitters are packaged in the same or different synaptic vesicles, and co-transmission as synaptic transmission occurring by multiple neurotransmitters and detected by distinct postsynaptic receptors (*Wallace and Sabatini, 2023*).

The cellular and molecular mechanisms of co-release are diverse across brain areas. In particular, the transmission modes of co-release, whether two neurotransmitters are released from the same vesicles or separate vesicles, and whether they are released from the same release sites or spatially segregated release sites, depend on the cell type in a given circuit. Co-packaging of GABA/glycine or glutamate/acetylcholine is found in the spinal cord (*Jonas et al., 1998*) and brainstem (*Nabekura et al., 2004*), and in the axon terminals of the medial habenula in the interpeduncular nucleus (*Ren et al., 2011*; *Frahm et al., 2015*), respectively. In contrast, the co-release of GABA and acetylcholine from different synaptic vesicles has been reported in the retina (*Lee et al., 2010*), frontal cortex (*Saunders et al., 2015*), and hippocampus (*Takács et al., 2018*). Furthermore, the spatially segregated co-release of glutamate/glycine or glutamate/dopamine has been reported in a subset of retinal amacrine cells (*Lee et al., 2016*), as well as in the axon terminals from a subset of midbrain dopamine neurons (*Zhang et al., 2015*; *Fortin et al., 2019*), respectively. Importantly, dual-transmitter co-release from distinct vesicle populations or release sites can result in different release properties (*Lee et al., 2010*; *Sengupta et al., 2017*; *Takács et al., 2018*; *Silm et al., 2019*; *Zych and Ford, 2022*) and induce diverse effects to spatially different postsynaptic target cells (*Lee et al., 2016*; *Sengupta et al., 2017*; *Granger et al., 2020*). These divergent transmission properties of co-release can regulate the circuit functions underlying distinct physiological and behavioral roles.

The co-release of glutamate and GABA, both fast-acting and functionally opposing neurotransmitters, has recently been identified in several brain regions of the adult brain (*Trudeau and El Mestikawy, 2018*; *Kim and Sabatini, 2022*; *Wallace and Sabatini, 2023*). In particular, the lateral habenula and dentate gyrus (DG) of the hippocampus have been extensively studied for the co-release of glutamate and GABA. The lateral habenula neurons receive glutamate/GABA co-releasing synaptic inputs from the entopeduncular nucleus (EP) (*Shabel et al., 2014*; *Wallace et al., 2017*; *Root et al., 2018*) and ventral tegmental area (VTA) (*Root et al., 2014*; *Yoo et al., 2016*; *Root et al., 2018*). Hypothalamic supramammillary nucleus (SuM) neurons projecting to the DG also co-release glutamate and GABA (*Pedersen et al., 2017*; *Hashimotodani et al., 2018*; *Billwiller et al., 2020*; *Chen et al., 2020*; *Li et al., 2020*; *Ajibola et al., 2021*). To understand how synaptic inputs mediated by the co-release of the excitatory action of glutamate and the inhibitory action of GABA are integrated and impact postsynaptic cells, it is necessary to elucidate how glutamate and GABA are co-released from the synaptic terminals. Mechanistically, if glutamate and GABA are co-released from distinct vesicle populations, each release could be differently regulated by distinct release mechanisms. Such distinct co-release modes of glutamate and GABA are expected to implement different presynaptic plasticity, which will have diverse effects on postsynaptic neurons in response to different activity patterns. By contrast, if two neurotransmitters are co-released from the same synaptic vesicles, both glutamate and GABA release are expected to be under the same regulation.

Whether glutamate and GABA are packaged in the same or separate vesicles remains controversial. Electrophysiological studies have demonstrated the co-packaging of glutamate and GABA in the same synaptic vesicles at EP-lateral habenular synapses (*Shabel et al., 2014*; *Kim et al., 2022*). Conversely, an immunoelectron microscopy (immuno-EM) study demonstrated that glutamatergic and GABAergic vesicle populations are segregated in the axon terminals of the EP and VTA projections in the lateral habenula, and that single terminals form both excitatory and inhibitory synapses (*Root et al., 2018*). The same study also reported that glutamate and GABA are segregated into distinct vesicle populations within the axon terminals of the SuM projections in the DG (*Root et al., 2018*). However, the mode of co-release of glutamate and GABA from the SuM terminals has never been functionally addressed.

In this study, we investigated the release properties of glutamate/GABA co-transmission at SuM-dentate granule cell (GC) synapses. We demonstrate that glutamate/GABA co-transmission shows independent synaptic responses and differential short-term plasticity and that their postsynaptic

targets are segregated. Our results suggest that the segregation of two different populations of transmitters makes it possible to exert short-term dynamic changes in the co-transmission balance of glutamate and GABA and modulate GC activity in a frequency-dependent manner.

## Results

### Glutamate/GABA co-transmission shows different release probabilities and Ca²⁺ sensitivities

To optogenetically stimulate SuM inputs in the DG, channelrhodopsin-2 (ChR2) was expressed in SuM neurons by injecting a Cre-dependent adeno-associated virus (AAV) encoding ChR2-eYFP (AAV-DIO-ChR2(H134R)-eYFP) into the lateral SuM of vesicular glutamate transporter 2 (VGluT2)-Cre mice (*Hashimotodani et al., 2018*; *Hirai et al., 2022*; *Tabuchi et al., 2022*; *Figure 1A and B*). Consistent with previous reports (*Boulland et al., 2009*; *Soussi et al., 2010*; *Root et al., 2018*; *Billwiller et al., 2020*; *Ajibola et al., 2021*), the vast majority of ChR2-eYFP-expressing SuM boutons in the DG co-expressed VGluT2 and vesicular GABA transporter (vesicular inhibitory amino acid transporter, VIAAT) (*Figure 1C–E*). We performed whole-cell patch-clamp recordings from GCs in acute hippocampal slices and recorded optically evoked excitatory or inhibitory postsynaptic currents (EPSCs or IPSCs) at SuM-GC synapses in the presence of picrotoxin or NBQX/D-AP5, respectively. To examine whether the co-release of glutamate and GABA exhibits the same or different release properties, we first monitored the paired-pulse ratio (PPR), an index of the change in presynaptic transmitter release (*Zucker and Regehr, 2002*), of glutamate/GABA co-transmission by paired stimulation (inter-stimulus interval: 100 ms). As reported previously (*Hashimotodani et al., 2018*; *Tabuchi et al., 2022*), both EPSCs and IPSCs at SuM-GC synapses exhibited paired-pulse depression (*Figure 1F*). Importantly, we found that PPR was significantly different between EPSCs and IPSCs; EPSCs were more depressed than IPSCs (*Figure 1G*; EPSC: 0.46 ± 0.03, n=8; IPSC: 0.6 ± 0.03, n=8, p<0.01, unpaired t test). It is possible that direct illumination of presynaptic terminals could increase transmitter release due to the broadening of the presynaptic waveform or direct influx of Ca²⁺ through ChR2 (*Jackman et al., 2014*; *Rost et al., 2022*). To avoid this possibility, we stimulated SuM axons instead of SuM terminals (*Figure 1—figure supplement 1A and B*), and obtained similar results to terminal illumination (*Figure 1—figure supplement 1C–F*). ChR2 desensitization could depress the second synaptic responses evoked by the paired-pulse stimulation due to the participation of fewer SuM fibers in mediating synaptic transmission. To test this possibility, the fiber volley evoked by light illumination was extracellularly recorded from supragranular layer. We found that the amplitudes of the second fiber volley was comparable to that of the first fiber volley (*Figure 1—figure supplement 2*), suggesting that ChR2-expressing SuM fibers were activated to the same extent by paired-pulse light illumination. Altogether, these results suggest that the release probability differs between the co-released glutamate and GABA from SuM terminals in the DG.

Since vesicular exocytosis depends on the extracellular Ca²⁺ concentration, we next examined whether reducing the release probability by lowering the extracellular Ca²⁺ concentration could have different effects on glutamate/GABA co-transmission. Reducing extracellular Ca²⁺ concentration from 2.5 to 1 mM decreased the amplitudes of EPSCs and IPSCs (*Figure 1H*; EPSC: n=14, p<0.001, paired t test; IPSC: n=15, p<0.001, paired t test). Remarkably, IPSCs decreased to a greater extent than EPSCs (*Figure 1I*; EPSC: 45.1 ± 4.1% of reduction, n=14; IPSC: 58.2 ± 2.7% of reduction, n=15, p<0.01, unpaired t test), suggesting different Ca²⁺ sensitivities for glutamate and GABA release.

### Co-release of glutamate and GABA differs in presynaptic modulation and Ca²⁺ channel-vesicle coupling configuration

To further investigate whether the release properties of glutamate and GABA at SuM-GC synapses are different, we next examined whether glutamate and GABA co-release are mediated by different types of Ca²⁺ channels. We found that the application of ω-conotoxin-GVIA (ω-CgTx), an N-type Ca²⁺ channel blocker, had no effect on EPSCs or IPSCs at SuM-GC synapses (*Figure 2A*; EPSC: 92.5 ± 13.8% of baseline, n=6, p=0.53, Wilcoxon signed-rank test; IPSC: 101.7 ± 11% of baseline, n=6, p=0.89, paired t test). We confirmed that ω-CgTx inhibited CA3-CA1 transmission (*Wu and Saggau, 1994*; *Figure 2—figure supplement 1*), suggesting that ω-CgTx we used is effective. In contrast, the application of ω-agatoxin-IVA (ω-Aga-IVA), a P/Q-type Ca²⁺ channel blocker, suppressed both EPSCs

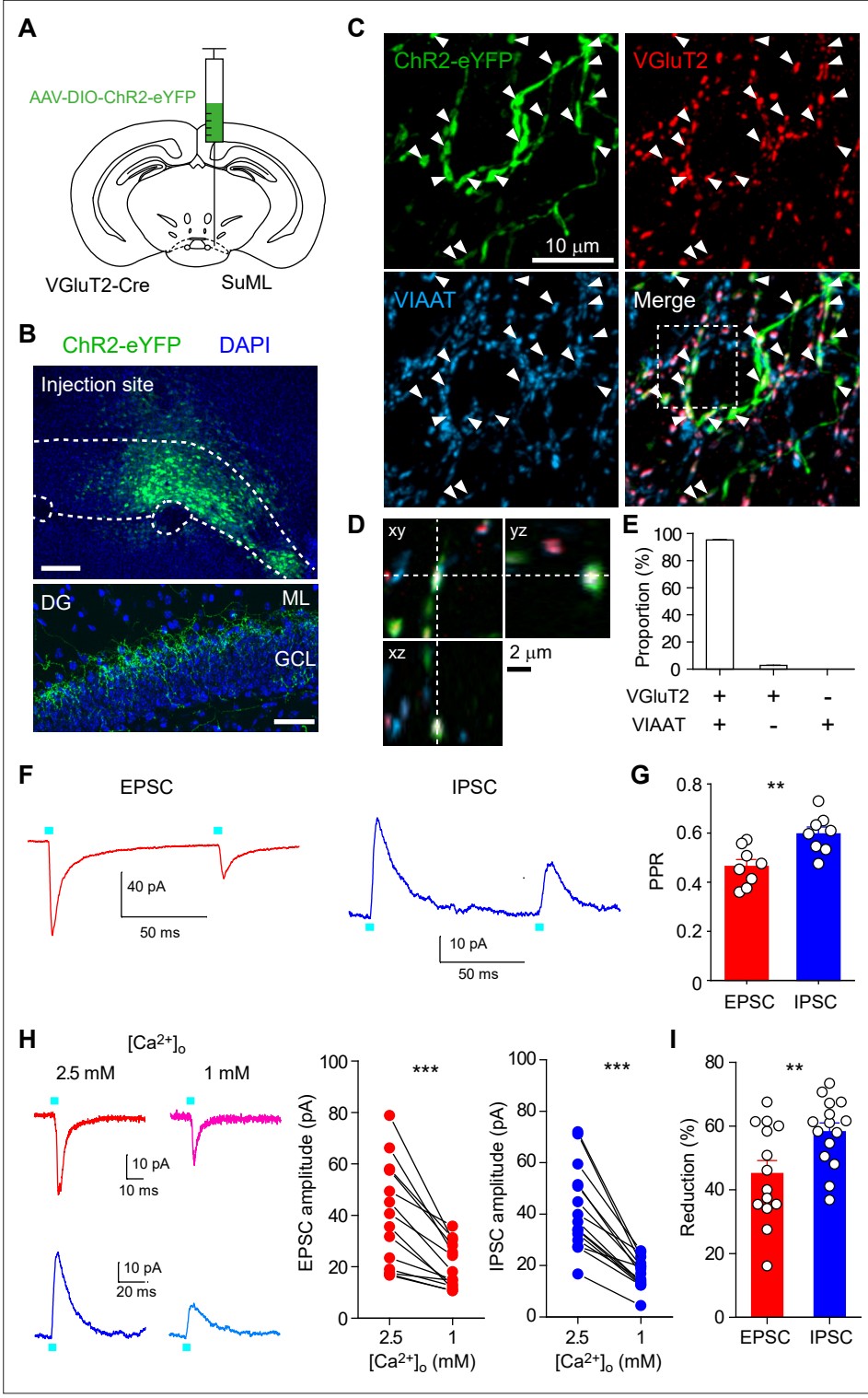

**Figure 1.** Different PPRs and Ca²⁺ sensitivities of glutamate/GABA co-transmission at SuM-GC synapses.
(**A**) Diagram illustrating the injection of AAV-DIO-ChR2(H134R)-eYFP into the lateral part of SuM of VGluT2-Cre mouse. (**B**) (top) Fluorescence image showing the injection site of AAV in the SuM. (bottom) ChR2(H134R)-eYFP-expressing SuM axons are observed in the supragranular layer of the DG. ML, molecular layer; GCL, granule cell layer. Scale bars, Top, 200 µm; Bottom, 50 µm.(**C**) Z-stacked immunofluorescence images double stained for VGluT2 (red) and VIAAT (cyan). The merged image demonstrates the colocalization of VGluT2 and VIAAT in the ChR2-eYFP-expressing SuM terminals (arrowheads). (**D**) Higher magnification xy, xz, and yz projection images

*Figure 1 continued on next page*

*Figure 1 continued*

outlined in boxed area in (**C**) show that a SuM bouton is co-stained with VGluT2 and VIAAT. (**E**) Proportion of VGluT2- and/or VIAAT-expressing boutons in the ChR2-eYFP labeled boutons (2205 boutons, 7 slices). Both VGluT2 and VIAAT (95.2 ± 0.3%), VGluT2 only (2.7 ± 0.2%), and VIAAT only (0%). (**F**) Representative traces of EPSC (red, $V_h$ = –70 mV) and IPSC (blue, $V_h$ = 0 mV) evoked with paired pulse illumination (100 ms interval). (**G**) Summary plot of PPR showing a significant difference between EPSC and IPSC. (**H**) Representative traces (left) and summary plots (right) of EPSCs and IPSCs in 2.5 and 1 mM extracellular $Ca^{2+}$. (**I**) Summary plot of percent reduction in the amplitudes of EPSCs and IPSCs from 2.5 to 1 mM extracellular $Ca^{2+}$. Data are presented as mean ± SEM. **p<0.01, ***p<0.001.

The online version of this article includes the following source data and figure supplement(s) for figure 1:

**Source data 1.** Source data displayed on *Figure 1*.

**Figure supplement 1.** PPRs of glutamatergic and GABAergic co-transmission show no difference between over-axon and over-bouton illumination.

**Figure supplement 1—source data 1.** Source data displayed on *Figure 1—figure supplement 1*.

**Figure supplement 2.** Paired-pulse light stimulation evoked similar amplitude of fiber volley.

**Figure supplement 2—source data 1.** Source data displayed on *Figure 1—figure supplement 2*.

and IPSCs to the same extent at SuM-GC synapses (*Figure 2B*; EPSC: 23.5 ± 4.6% of baseline, n=5, p<0.001, paired t test; IPSC: 23.9 ± 1.9% of baseline, n=6, p<0.05, Wilcoxon signed-rank test; EPSC vs IPSC: p=0.92, unpaired t test), suggesting that both glutamate and GABA release rely on P/Q-type $Ca^{2+}$ channels at SuM-GC synapses.

If the synaptic vesicle pools for glutamate and GABA are identical, the effects of presynaptic G-protein-coupled receptor (GPCR) modulation on both transmissions are expected to be the same. We therefore examined how glutamate/GABA co-transmission is regulated by presynaptic GPCRs. We previously demonstrated that the co-transmission of glutamate and GABA at SuM-GC synapses is presynaptically suppressed by group II metabotropic glutamate receptors (mGluRs) and GABA$_B$ receptors (*Hashimotodani et al., 2018*). In agreement with this, application of the group II mGluR agonist DCG-IV suppressed both EPSCs and IPSCs at SuM-GC synapses (*Figure 2C*). Notably, DCG-IV inhibited EPSCs (IC$_{50}$=0.11 μM) to a greater extent than it did IPSCs (IC$_{50}$=0.27 μM; *Figure 2C*; two-way ANOVA, $F_{1,51}$ = 8.1, p<0.01). Similar results were found with the application of 5 μM GABA$_B$ receptor agonist baclofen, which suppressed EPSCs more than IPSCs (*Figure 2D*; EPSC: 51.6 ± 2.4% inhibition, n=8; IPSC: 38.6 ± 4.3% inhibition, n=8, p<0.05, unpaired t test). These differential inhibitory effects of GPCRs on glutamatergic and GABAergic co-transmission are contrary to the expected idea of identical vesicle pools for glutamate and GABA but favor the hypothesis of separate vesicles for glutamate and GABA.

Spatial coupling between presynaptic $Ca^{2+}$ channels and $Ca^{2+}$ sensors of exocytosis is an essential feature in determining the properties of neurotransmitter release, and tight or loose coupling is distinguished by using two different binding rate $Ca^{2+}$ chelators BAPTA and EGTA (*Adler et al., 1991*; *Eggermann et al., 2011*). We examined the effects of these two $Ca^{2+}$ chelators on the co-transmission of glutamate and GABA. We found that the bath application of BAPTA-AM suppressed both EPSCs and IPSCs to the same extent (EPSC: 63.4 ± 12.2% of baseline, n=7, p<0.01, paired t test; IPSC: 63.4 ± 7.8% of baseline, n=8, p<0.001, paired t test; EPSC vs. IPSC: p=0.96, unpaired t test; *Figure 2E*). In contrast, EGTA-AM suppressed EPSCs, whereas it did not affect IPSCs (EPSC: 67.4 ± 7.2% of baseline, n=6, p<0.01, paired t test; IPSC: 93.1 ± 9.2% of baseline, n=6, p=0.40, Wilcoxon signed-rank test; EPSC vs. IPSC: p<0.05, unpaired t test; *Figure 2E*). These results indicate tighter coupling between $Ca^{2+}$ channels and synaptic vesicles for GABA release than for glutamate release.

## Minimal light stimulation of SuM inputs elicits independent EPSCs and IPSCs in GCs

Given that light stimulation at normal intensity elicits synchronized release of neurotransmitters from several SuM terminals, it is difficult to evoke a synaptic response mediated by a single vesicle release. For this purpose, we next performed minimal light stimulation to detect stochastic synaptic responses of independent EPSCs and IPSCs, or compound EPSC/IPSC responses (*Kim et al., 2022*). In this experiment, optical stimulation (over-bouton illumination) was delivered in the presence

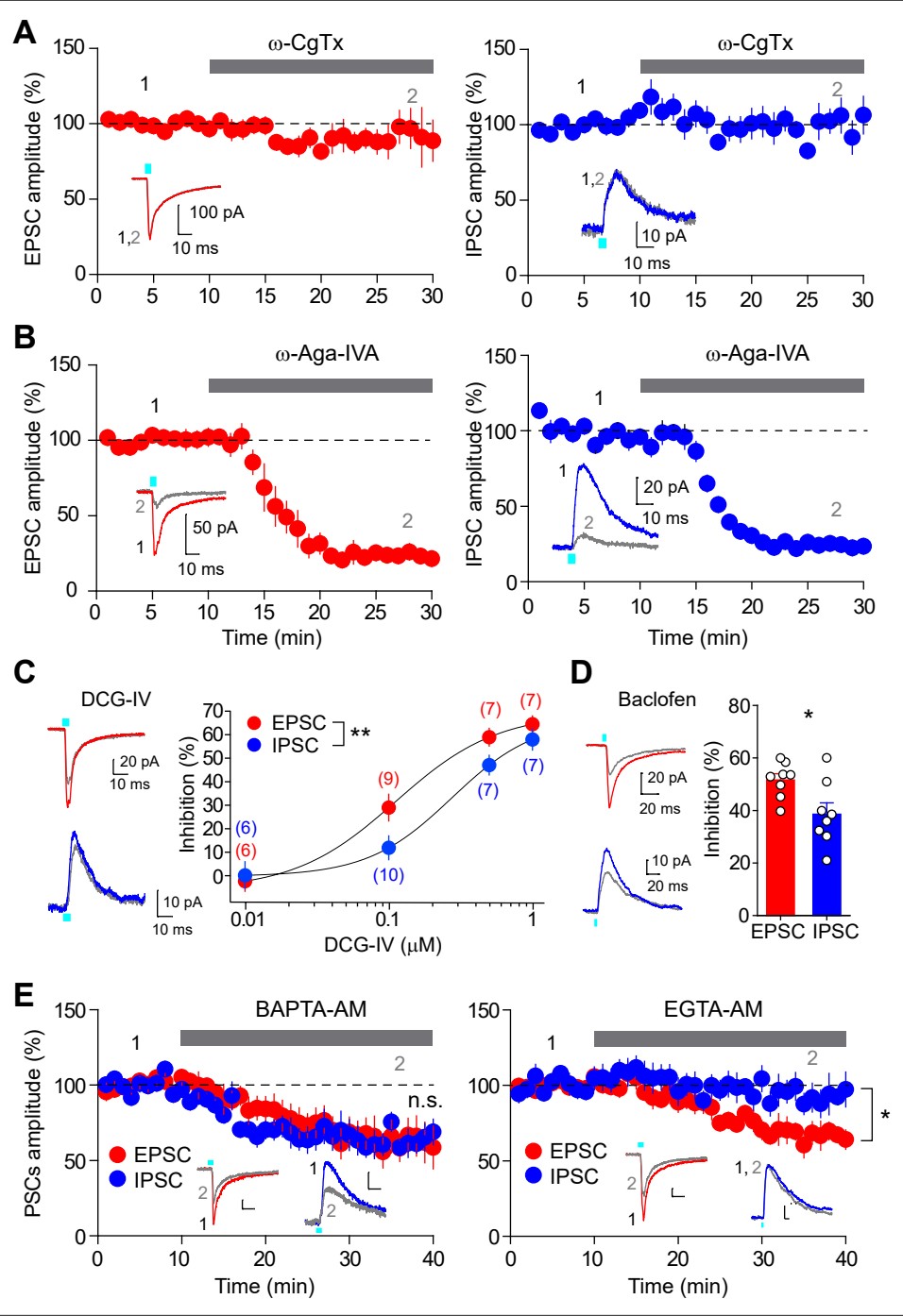

**Figure 2.** Different presynaptic modulation and Ca²⁺ chelator-sensitivities of glutamate/GABA co-transmission.
(**A**) Time course summary plots showing the effects of ω-CgTx (500 nM) on co-transmission of glutamate (left) and GABA (right) at SuM-GC synapses. Insets indicate representative traces. (**B**) Time course summary plots showing that the co-transmission of glutamate (left) and GABA (right) at SuM-GC synapses were inhibited by ω-Aga-IVA (200 nM). Insets show representative traces. (**C**) (left) Representative traces before (EPSC, red; IPSC, blue) and after (gray) application of DCG-IV (0.1 μM). (right) Summary plot of concentration-response curves. Data are fitted to the Hill equation. Numbers in parentheses indicate the number of cells. (**D**) (left) Representative traces before (EPSC, red; IPSC, blue) and after (gray) application of baclofen (5 μM). (right) Summary plot of percent inhibition in the amplitudes of EPSCs and IPSCs by 5 μM baclofen. (**E**) Time course summary plots showing the sensitivity of EPSCs and IPSCs to 100 μM BAPTA-AM (left) and 100 μM EGTA-AM (right). Insets show representative traces. Calibration: 10 pA, 10 ms. Data are presented as mean ± SEM. *p<0.05, **p<0.01. n.s., not significant.

*Figure 2 continued on next page*

*Figure 2 continued*

The online version of this article includes the following source data and figure supplement(s) for figure 2:

**Source data 1.** Source data displayed on *Figure 2*.

**Figure supplement 1.** The effect of $\omega$-CgTx on CA3-CA1 transmission.

**Figure supplement 1—source data 1.** Source data displayed on *Figure 2—figure supplement 1*.

of tetrodotoxin (TTX) and 4-aminopyridine (4-AP) to exclude SuM inputs-mediated polysynaptic responses, together with D-AP5 to block NMDA receptors. We also decreased the extracellular $Ca^{2+}$ concentration to 1 mM to reduce the release probability. At intermediate holding membrane potentials (–20 to –30 mV), maximum light power illumination elicited biphasic responses (EPSC-IPSC sequence), enabling simultaneous recordings of glutamate/GABA co-transmission in GCs (*Figure 3A*, left). After confirming the existence of biphasic PSCs, the light power intensity was decreased to detect small synaptic events, that were stochastically observed among the 100 times stimulations (success rate=~15%; *Figure 3A, B*, *Figure 3—figure supplement 1*). Using this method, we analyzed 1700 trials from 17 cells and detected 86 EPSC-only, 67 IPSC-only, and 19 biphasic events (*Figure 3B and C*). These results indicate that independent EPSCs and IPSCs are the majority of synaptic responses (*Figure 3D*; EPSC: 5.1 ± 0.5% probability; IPSC: 3.9 ± 0.4% probability; biphasic: 1.1 ± 0.3% probability, n=17, EPSC vs biphasic, p<0.001; IPSC vs biphasic, p<0.001, one-way ANOVA with Tukey's *post hoc* test). To exclude the possibility that the independent EPSCs and IPSCs detected by minimal light stimulation might have masked the putative tiny synaptic responses of their counterpart currents mediated by the associated transmitters, the amplitudes and kinetics of minimal stimulation-evoked EPSCs and IPSCs were compared with or without blockers of their counterpart receptors. We found that the amplitudes and decay of minimal light stimulation-evoked EPSCs were not changed by the application of the $GABA_A$ receptor blocker picrotoxin (*Figure 3—figure supplement 2*). Similarly, the amplitudes and rise time of minimal light stimulation-evoked IPSCs were not altered by application of the AMPA/kainate receptor blocker NBQX (*Figure 3—figure supplement 2*). These results validate that EPSCs and IPSCs evoked by minimal optical stimulation are mediated by the stochastic release of glutamate and GABA, respectively, from single vesicles. Furthermore, by comparing minimal stimulation-evoked PSCs with strontium-induced asynchronous PSCs (see below), we found that the amplitudes of EPSCs and IPSCs evoked by minimal stimulation were identical to those of strontium-induced asynchronous EPSCs and IPSCs, respectively (*Figure 3G*), suggesting that PSCs evoked by minimal stimulation were mediated by a single vesicle release.

Taken together, these results indicate that glutamatergic and GABAergic co-transmission occurs independently at SuM-GC synapses.

## Asynchronous release from SuM terminals exhibits independent EPSCs and IPSCs in GCs

It should be noted that minimal light stimulation of SuM inputs may occasionally elicit synchronized release of glutamate and GABA from different vesicle pools, possibly due to the simultaneous opening of several $Ca^{2+}$ channels, which could account for the substantial biphasic responses we detected (*Figure 3D*). To minimize the likelihood of synchronized release, we recorded asynchronous quantal PSCs by substitution of extracellular $Ca^{2+}$ with $Sr^{2+}$. At intermediate holding membrane potentials (–20 to –30 mV), light pulses elicited synchronized biphasic responses in the extracellular $Ca^{2+}$ (*Figure 3E*). By replacing extracellular $Ca^{2+}$ with $Sr^{2+}$, asynchronous quantal synaptic responses were recorded following light stimulation of the SuM inputs (*Figure 3E*). By repeating the light stimulation of the SuM inputs 30 times (every 10 s), we found that the majority of asynchronous synaptic events were EPSC-only or IPSC-only, whereas biphasic PSCs represented only a minor population (*Figure 3F*; EPSC: 27.9 ± 4.7 events; IPSC: 23.4 ± 4.2 events; biphasic: 2.5 ± 0.5 events, n=11, EPSC vs biphasic, p<0.001; IPSC vs biphasic, p<0.001, one-way ANOVA with Tukey's *post hoc* test). These results further support our conclusion that glutamate/GABA co-transmission is mediated by glutamate and GABA co-released from different populations of synaptic vesicles.

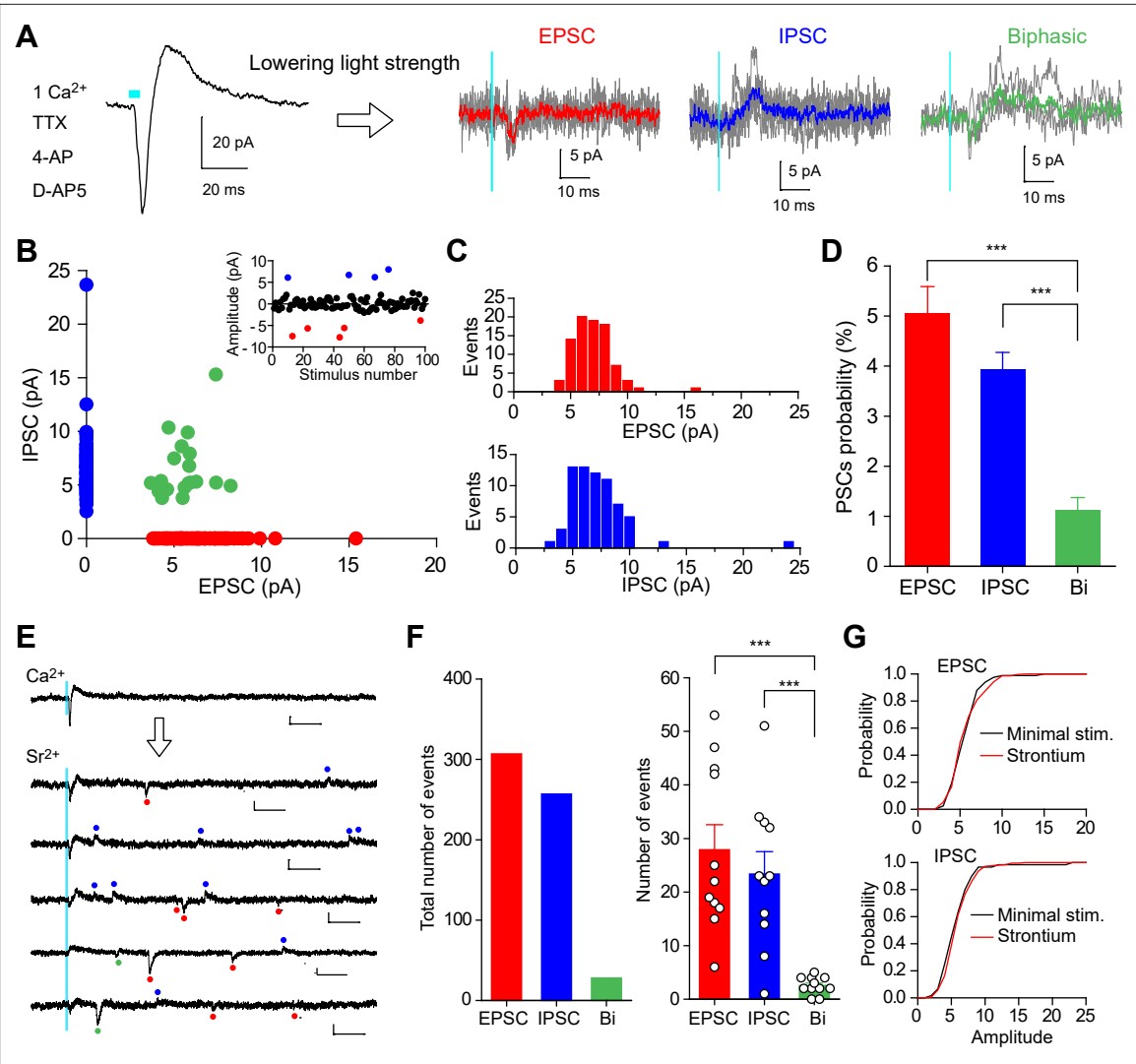

**Figure 3.** Uniquantal release-mediated synaptic responses evoked by minimal light stimulation or strontium-induced asynchronous release at SuM-GC synapses. (**A**) Representative traces of biphasic response elicited by light stimulation with maximum light power (left) and stochastically evoked PSCs with minimal light stimulation (100 trials for each cell) (right). Synaptic responses at SuM-GC synapses were recorded at holding potentials of –20 to –30 mV in the presence of TTX (1 µM), 4-AP (1 mM), D-AP5 (50 µM), and $Ca^{2+}$ (1 mM). Average EPSC (red), IPSC (blue), and biphasic current (green) are superimposed on individual traces. (**B**) Scatter plot of the amplitude of IPSCs against the amplitude of EPSCs recorded from 17 cells. Success events of 172 PSCs are plotted. (Inset) A single experiment showing 100 trials with an interval of 10 sec elicited stochastically EPSCs (red) and IPSCs (blue). (**C**) Amplitude histograms of EPSCs (top) and IPSCs (bottom) in same data as in B. (**D**) Summary graph of PSCs probability. (**E**) Representative traces showing strontium-induced asynchronous release recorded from GC at holding potentials of –20 to –30 mV. The NMDA receptor antagonist D-AP5 (50 µM) was included in the extracellular solution to eliminate NMDA receptor-mediated EPSCs. Red, blue, and green points indicate detected EPSCs, IPSCs, and biphasic currents, respectively. Calibration: 10 pA, 50 ms. (**F**) (left) Summary bar graph showing the total number of asynchronous quantal responses (n=11, 30 trials for each cell; EPSC: 307 events; IPSC: 257 events; biphasic: 28 events). (right) Averaged number of asynchronous events showing that the majority of asynchronous events are EPSCs and IPSCs, and a few biphasic responses. (**G**) Cumulative distributions of the amplitudes of EPSCs (top) and IPSCs (bottom) for minimal light stimulation-evoked PSCs and strontium-induced PSCs (EPSC: minimal stim.: 86 events from 17 cells; strontium: 307 events from 11 cells, p=0.72; IPSC: minimal stim.: 67 events from 17 cells; strontium: 256 events from 11 cells, p=0.096, Kolmogorov-Smirnov test). Data are presented as mean ± SEM. ***p<0.001.

The online version of this article includes the following source data and figure supplement(s) for figure 3:

**Source data 1.** Source data displayed on *Figure 3*.

**Figure supplement 1.** Minimal light stimulation-evoked synaptic responses at SuM-GC synapses.

**Figure supplement 2.** The amplitudes and kinetics of minimal light stimulation-evoked EPSCs or IPSCs at SuM-GC synapses were not altered by blockade of their counterpart currents.

**Figure supplement 2—source data 1.** Source data displayed on *Figure 3—figure supplement 2*.

## Segregated glutamatergic and GABAergic postsynaptic sites at SuM-GC synapses

We then investigated whether SuM-GC synapses exhibit distinct postsynaptic sites for each neurotransmitter using immunohistochemistry. To enhance spatial resolution, we used ultrathin sections (100-nm-thick) for analysis. As shown previously (*Uchigashima et al., 2011*), double immuno-fluorescence revealed that almost all VGluT2-positive terminals in the GC layer co-expressed VIAAT (*Figure 4A and B*). Further post-embedding immunogold EM (*Figure 4C and D*) showed that these terminals formed asymmetric and symmetric synapses, characteristic of excitatory and inhibitory synapses, respectively (*Figure 4C and D*), confirming previous observations (*Boulland et al., 2009*; *Soussi et al., 2010*; *Root et al., 2018*; *Billwiller et al., 2020*). Conventional EM using serial ultrathin sections also revealed that a single nerve terminal formed both asymmetric and symmetric synapses on a GC soma (*Figure 4E*). The formation of asymmetric and symmetric synapses by a single SuM terminal was further confirmed by pre-embedding immunogold EM for eYFP, which was expressed in the SuM neurons by injection of AAV-DIO-ChR2(H134R)-eYFP into the SuM of VGluT2-Cre mice (*Figure 4F*). It has been reported that VTA neurons also project to the GC layer of the DG and co-release glutamate and GABA (*Ntamati and Lüscher, 2016*). However, our retrograde tracing with fluorescent micro-spheres injected into the dorsal DG did not reveal any retrogradely labeled neurons in the VTA, while labeled neurons were successfully observed in the SuM and entorhinal cortex (EC) (*Figure 4—figure supplement 1*). This suggests that most of the VGluT2-positive terminals (co-expressed with VIAAT, *Figure 4A and B*) in the GC layer most likely originated from the SuM. To examine the spatial asso-ciation of SuM terminals with glutamatergic and GABAergic postsynaptic sites, we performed triple immunofluorescence for VGluT2, the NMDA receptor subunit GluN1, and GABAergic postsynaptic sites by the GABA$_A$ receptor subunit GABA$_A\alpha$1. In the GC layer, almost all VGluT2-positive terminals were associated with GluN1 and/or GABA$_A\alpha$1 (*Figure 4G and H*). Specifically, the majority of VGluT2-positive terminals were associated with both GluN1 and GABA$_A\alpha$1 (52.5%), and the remainder were associated with GABA$_A\alpha$1 (22.1%) alone or GluN1 (25.4%) alone (*Figure 4I*). To investigate the spatial relationship between glutamatergic and GABAergic synapses, we quantified the distance between GluN1 and GABA$_A\alpha$1 cluster adjacent to the same VGluT2-positive terminal. On average, the distance was 397.5 ± 8.75 nm, and most VGluT2-positive terminals (79.2%, 191 out of 241) form glutamatergic and GABAergic synapses within 500 nm (*Figure 4J*). Taken together, these results suggest that a single SuM terminal forms GluN1-containing excitatory synapse and GABA$_A\alpha$1-containing inhibitory synapse in separate locations, yet they remain closely proximal.

## Glutamate/GABA co-transmission balance of SuM inputs is dynamically changed in a frequency-dependent manner

The results described thus far indicate that the co-transmission of glutamate and GABA is spatially segregated at SuM-GC synapses, thereby making it possible to independently regulate the co-release of glutamate and GABA from the same SuM terminals. Due to the different release properties of glutamate and GABA in the SuM terminals, glutamate/GABA co-transmission could exhibit different patterns of short-term plasticity at SuM-GC synapses. To test this possibility, we delivered a train of 10 light stimuli at 5–20 Hz, physiologically relevant frequencies (*Kirk and McNaughton, 1991*; *Kocsis and Vertes, 1994*; *Ito et al., 2018*; *Farrell et al., 2021*), to ChR2-expressing SuM inputs, and compared the short-term depression of EPSCs with that of IPSCs (*Figure 5A*). To exclude the puta-tive involvement of pre- and/or postsynaptic modulations by mGluRs and GABA$_B$ receptors during a train of stimulation, pharmacologically isolated EPSCs or IPSCs were recorded in the presence of the broad-spectrum mGluR antagonist LY341495 (100 µM) and GABA$_B$ receptor antagonist CGP55845 (3 µM). We found that trains of 10 light stimuli at 5, 10, and 20 Hz elicited pronounced short-term synaptic depression of EPSCs during train with stronger depression at higher frequencies (*Figure 5A and B*). We further performed the same experiments under recording conditions with a low release probability by reducing the extracellular Ca$^{2+}$ concentration from 2.5 to 1 mM. While train stimulation elicited a weaker depression than 2.5 mM extracellular Ca$^{2+}$, frequency-dependent depression was still observed (*Figure 5C*). In marked contrast to EPSCs, while IPSCs also exhibited short-term synaptic depression across all tested frequencies, GABAergic co-transmission showed a similar magnitude of depression across all frequencies (*Figure 5D and E*). Similarly, short-term depression of IPSCs did not show a frequency-dependent depression with 1 mM extracellular Ca$^{2+}$ (*Figure 5F*), leading to

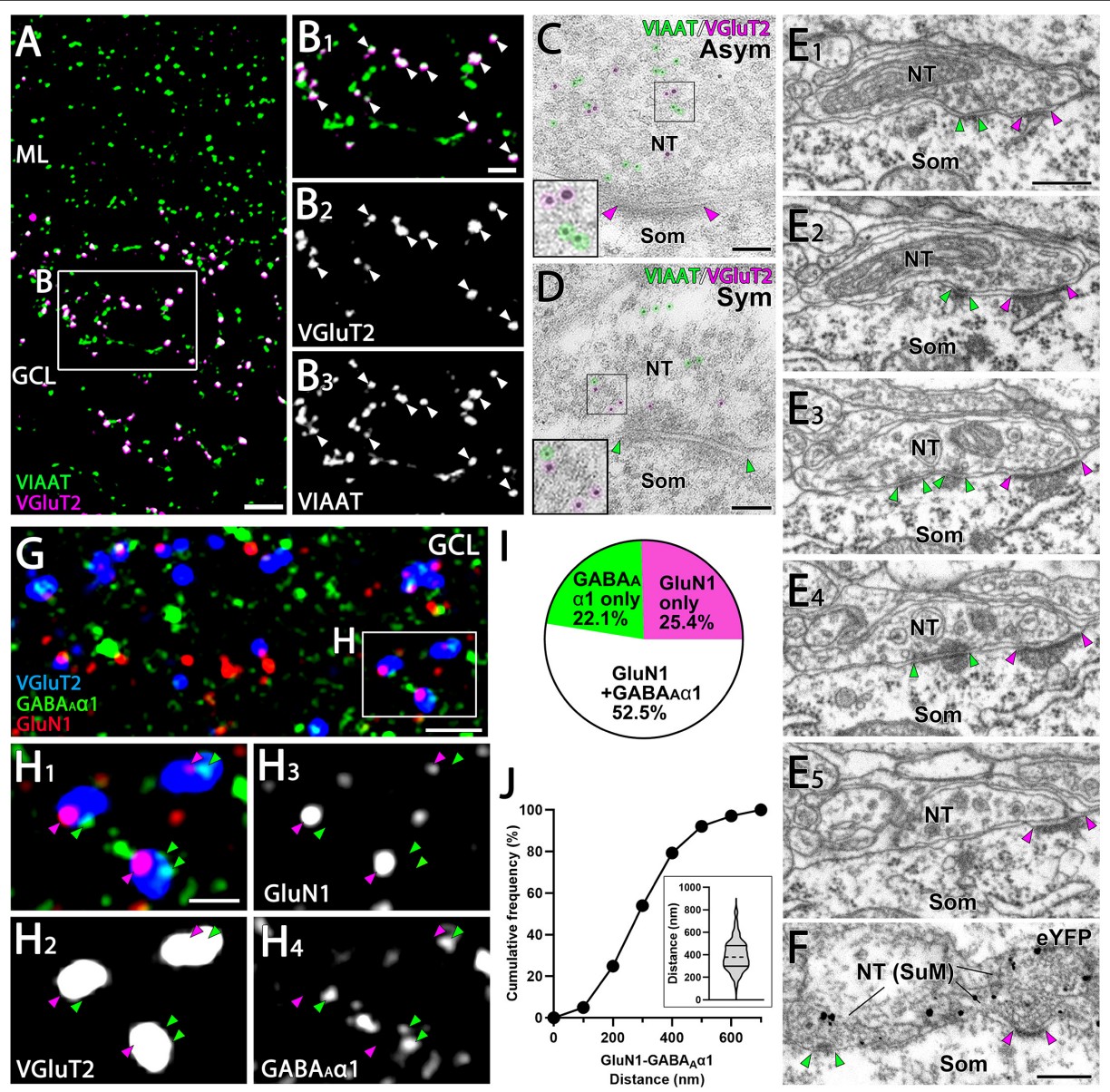

**Figure 4.** Close association of GluN1 and GABA$_A$α1 facing the identical SuM terminals. (**A, B**) Double immunofluorescence for VIAAT (green) and VGluT2 (magenta) in the GC layer of the DG. (**B**) Higher magnification images of the boxed area in (**A**) show that VGluT2-positive terminals are consistently co-labeled with VIAAT (white arrowheads). (**C, D**) Post-embedding immunogold EM shows immunogold particles for VIAAT (green, 5 nm) and VGluT2 (magenta, 10 nm) colocalizing within the same terminal, forming asymmetric (**C**) and symmetric (**D**) synapses. NT, nerve terminal; Som, soma. (**E**) Five consecutive EM images depict a single NT forming both asymmetric and symmetric synapses with a GC soma. The postsynaptic densities of the asymmetric and symmetric synapses are indicated by magenta and green arrowheads, respectively. (**F**) A pre-embedding immunogold EM image demonstrates a single NT from SuM, labeled with eYFP, forming both asymmetric and symmetric synapses with a GC soma. The postsynaptic densities are similarly marked by magenta and green arrowheads. (**G, H**) Triple immunofluorescence for GluN1 (red), GABA$_A$α1 (green), and VGluT2 (blue) in the GC layer of the DG. (**H**) Higher magnification of the boxed region in (**G**). Magenta arrowheads indicate GluN1 and green arrowheads indicate GABA$_A$α1, both apposed to VGluT2 puncta. (**I**) The proportion of VGluT2-positive terminals in the GC layer associated with GluN1 *and/or* GABA$_A$α1 immunoreactivity. Data are based on measurements from 634 VGluT2-positive terminals from two mice. (**J**) Cumulative distribution of the distance between GluN1 and GABA$_A$α1 puncta associated with VGluT2-positive terminals in the GC layer that are double-labeled for GluN1 and GABA$_A$α1. Data are based on measurements from 241 VGluT2-positive terminals from two mice (violin plot shown in the inset). The dashed line within each violin plot represents the median, while the solid lines at the top and bottom indicate the 75th and 25th percentiles, respectively. Scale bars: **A**, 5 µm; **B**, 2 µm; **C, D**, 100 nm; **E, F**, 400 nm; **G**, 5 µm; **H**, 1 µm.

The online version of this article includes the following source data and figure supplement(s) for figure 4:

**Source data 1.** Source data displayed on *Figure 4*.

*Figure 4 continued on next page*

*Figure 4 continued*

**Figure supplement 1.** DG receives monosynaptic input from SuM and EC, but not from VTA.

a sustained inhibitory effect at any frequency. Given that EPSCs show strong frequency-dependent depression, whereas IPSCs exhibit frequency-independent sustained depression, it is likely that the EPSC/IPSC ratio can decrease during high-frequency train stimulation. Indeed, comparing the amplitudes of the 10th EPSCs with those of the 10th IPSCs indicated that the depression of EPSCs was stronger than that of IPSCs at 10 Hz and 20 Hz (*Figure 5G*). We further addressed whether short-term dynamic changes in the EPSC/IPSC ratio could be observed in the same cell by recording the biphasic

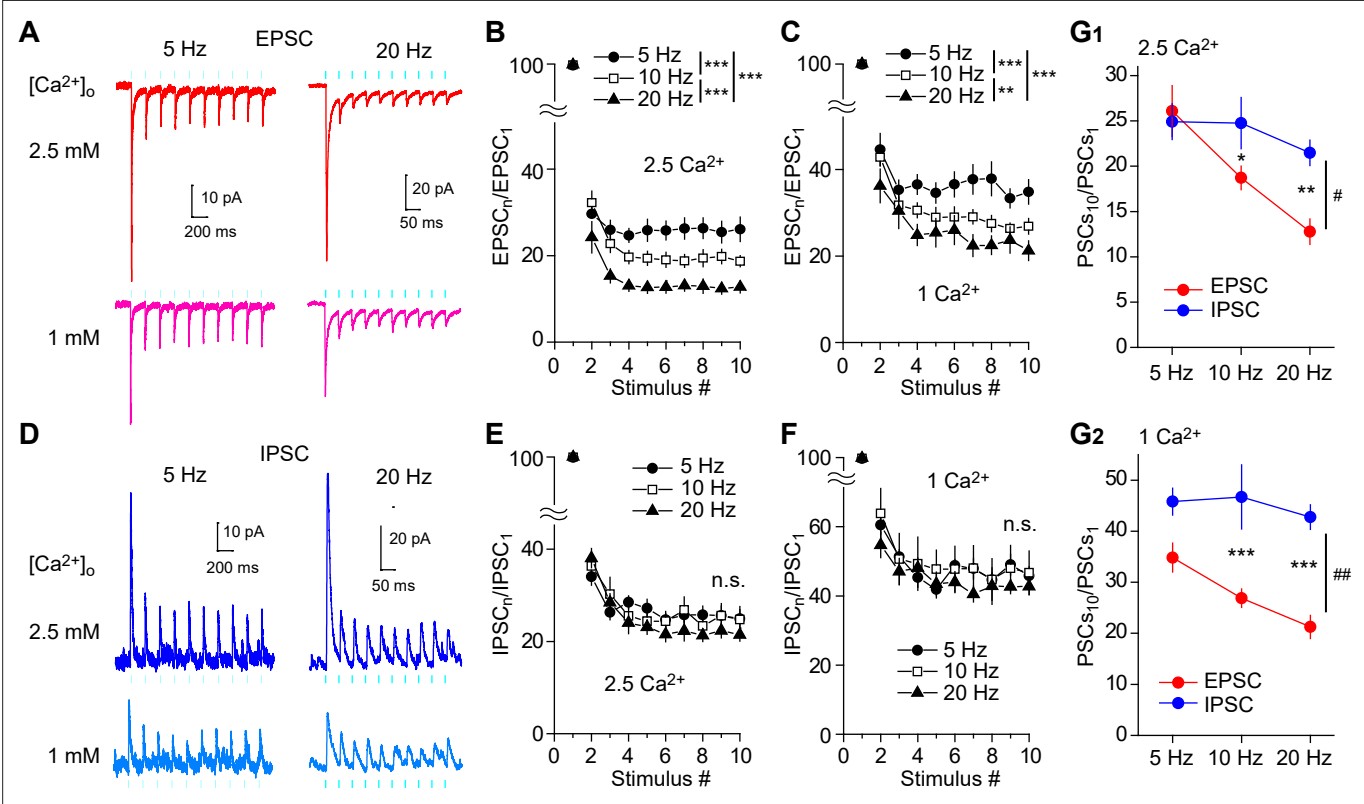

**Figure 5.** Frequency-dependent shift of glutamate/GABA co-transmission balance of SuM inputs in GCs. (**A, D**) Representative traces of EPSCs ($V_h$ = –70 mV) (**A**) and IPSCs ($V_h$ = 0 mV) (**D**) in response to 10 light stimuli at 5 Hz (left) or 20 Hz (right) in 2.5 mM (top) or 1 mM (bottom) extracellular $Ca^{2+}$. (**B**) Summary graph of normalized EPSC amplitude plotted against the stimulus number in 2.5 mM extracellular $Ca^{2+}$. Two-way repeated measures ANOVA, $F_{(2,16)}$ = 13.0, p<0.001, n=9 or 10; Tukey post hoc test: ***p<0.001. (**C**) Same as (**B**), but recorded in 1 mM extracellular $Ca^{2+}$. Two-way repeated measures ANOVA, $F_{(2,16)}$ = 11.4, p<0.001, n=9 or 10; Tukey post hoc test: **p<0.01, ***p<0.001. (**E**) Summary graph of normalized IPSC amplitude plotted against stimulus number in 2.5 mM extracellular $Ca^{2+}$. Two-way repeated measures ANOVA, $F_{(2,16)}$ = 0.003, p=0.997, n=9. (**F**) Same as (**E**), but recorded in 1 mM extracellular $Ca^{2+}$. Two-way repeated measures ANOVA, $F_{(2,16)}$ = 0.02, p=0.981, n=9. (**G**) Summary plots showing the normalized amplitudes of 10th EPSCs and IPSCs at 5 Hz, 10 Hz, and 20 Hz in 2.5 mM extracellular $Ca^{2+}$ (G1: two-way repeated measures ANOVA, $F_{(1,7)}$ = 8.03, #p<0.05, n=9 or 10; Tukey's post hoc test, EPSC versus IPSC, *p<0.05, **p<0.01), or in 1 mM extracellular $Ca^{2+}$ (G2: two-way repeated measures ANOVA, $F_{(1,7)}$ = 21.76, ##p<0.01, n=9 or 10; Tukey's post hoc test, EPSC versus IPSC, ***p<0.001). n.s., not significant. Data are presented as mean ± SEM.

The online version of this article includes the following source data and figure supplement(s) for figure 5:

**Source data 1.** Source data displayed on *Figure 5*.

**Figure supplement 1.** Short-term changes in EPSC/IPSC ratio during train stimulation at SuM-GC synapses.

**Figure supplement 1—source data 1.** Source data displayed on *Figure 5—figure supplement 1*.

**Figure supplement 2.** High fidelity activation of ChR2-expressing SuM fibers during trains.

**Figure supplement 2—source data 1.** Source data displayed on *Figure 5—figure supplement 2*.

**Figure supplement 3.** Preventing postsynaptic saturation and desensitization does not alter short-term depression of EPSCs and IPSCs.

**Figure supplement 3—source data 1.** Source data displayed on *Figure 5—figure supplement 3*.

synaptic responses at intermediate membrane potentials that enabled simultaneous monitoring of the compound EPSC/IPSC. We found that 10 Hz train light illumination induced a stronger depression of EPSCs than of IPSCs (*Figure 5—figure supplement 1*), thereby decreasing the EPSC/IPSC ratio during the high-frequency train stimulation. We confirmed that the fiber volley induced by trains of 10 light stimuli was comparable across all tested frequencies (*Figure 5—figure supplement 2*), indicating that a similar number of SuM fibers were activated with each light pulse during trains.

Despite the differential properties of the co-release of glutamate and GABA from SuM terminals, the difference in the magnitude of short-term depression between glutamatergic and GABAergic co-transmission could be mediated by postsynaptic mechanisms, such as AMPA receptor saturation and desensitization during burst high-frequency stimulation (*Trussell et al., 1993*; *Brenowitz and Trussell, 2001*; *Heine et al., 2008*). To test this possibility, we recorded EPSC trains in the presence of γ-D-glutamylglycine (γDGG), a low-affinity AMPA receptor antagonist that relieves AMPA receptor saturation and desensitization (*Wadiche and Jahr, 2001*; *Wong et al., 2003*; *Foster and Regehr, 2004*). As expected, 2 mM γDGG reduced the amplitude of EPSCs (60.1 ± 4.3% reduction, n=16, p<0.001, paired t test). Under these conditions, short-term depression with high-frequency light stimulation (10 Hz and 20 Hz) was not affected (*Figure 5—figure supplement 3A, B*). This rules out a postsynaptic contribution to the strong depression of EPSCs and suggests that presynaptic mechanisms mediate this short-term depression. Similarly, to determine whether postsynaptic GABA$_A$ receptor saturation is involved in short-term depression during train stimulation, we used the low-affinity competitive GABA$_A$ receptor antagonist TPMPA (*Sakaba, 2008*; *Markwardt et al., 2009*; *Turecek et al., 2016*). We found that 300 µM TPMPA reduced the amplitude of IPSCs (47.1 ± 3.8% reduction, n=11, p<0.01, Wilcoxon signed-rank test), while short-term depression of IPSCs in response to 10 light pulses at 10 Hz and 20 Hz did not differ before and after the application of TPMPA (*Figure 5—figure supplement 3C, D*), excluding the involvement of GABA$_A$ receptor saturation in the short-term depression of IPSCs at SuM-GC synapses, suggesting a presynaptic origin of this short-term plasticity.

Taken together, these results suggest that SuM-GC synapses exhibit short-term dynamic changes in the glutamate/GABA co-transmission ratio during repetitive high-frequency SuM activity due to the different properties of frequency-dependent depression between glutamatergic and GABAergic transmission, favoring inhibition over excitation. Furthermore, the different properties of short-term depression of EPSCs and IPSCs also support the segregated co-transmission of glutamate and GABA at SuM-GC synapses.

## SuM inputs modulate GC activity in a frequency-dependent manner

We have previously demonstrated that SuM inputs associated with EC inputs facilitate GC firing (*Hashimotodani et al., 2018*). Our previous study investigated the single association of synaptic inputs, but did not examine repetitive stimulation of synaptic inputs. We investigated how the repetitive activation of SuM inputs, which implement frequency-dependent filtering of glutamatergic responses with sustained synaptic inhibition by GABAergic co-transmission, affects GC excitability. To mimic excitatory EC synaptic inputs onto GCs, a sinusoidal current was injected into the soma of GCs (*Kamondi et al., 1998*; *Magee, 2001*; *Ajibola et al., 2021*), which can bypass direct synaptic stimulation of the perforant path (main excitatory inputs originating from EC) and, consequently, circumvent the recruitment of inhibitory circuitry. We found that a sinusoidal current at 5 Hz generated action potentials at the peak of the depolarizing phase (*Figure 6A*). The injected current was adjusted to generate action potentials with fewer than five spikes in each trial (10 cycles), and the optogenetically activated SuM inputs were paired at the depolarizing phase of each cycle. Paired light illumination significantly enhanced GC firing compared to sinusoidal current injections without light stimulation throughout the trial (10 stimuli; *Figure 6A and D*). Notably, the probability of spikes further increased when GABAergic transmission was blocked by picrotoxin (*Figure 6A and D*). These results suggest that SuM inputs enhance GC firing and that GABAergic co-transmission negatively regulates the net excitatory effect of SuM inputs at 5 Hz, similar to the effects of a single associational activation of SuM and EC inputs (*Hashimotodani et al., 2018*). In striking contrast, at higher frequencies (10 Hz and 20 Hz), the enhancement of spike probability by SuM activation was attenuated and disappeared in the last few stimulations (*Figure 6B, C, E and F*). Importantly, in the presence of picrotoxin, SuM activation by light stimulation enhanced GC firing, even in the last few stimulations (*Figure 6B, C, E and F*), suggesting that SuM glutamatergic co-transmission still exerted excitatory effects in the

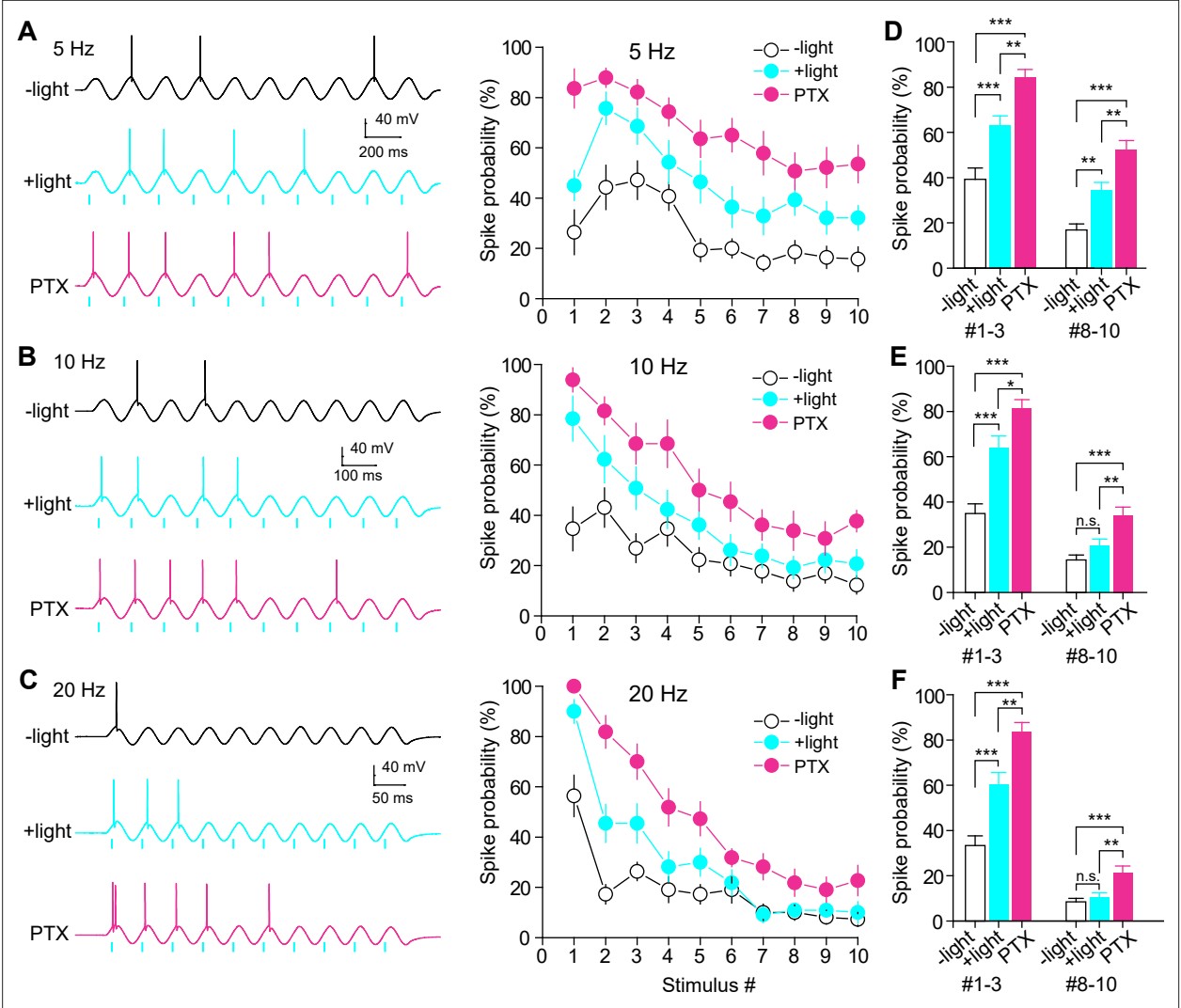

**Figure 6.** Frequency-dependent modulation of GC firing by SuM inputs. (A–C) (left) Representative traces showing GC spikes in response to sinusoidal current injections without (top) and with (middle) paired light stimulation of SuM inputs at 5 Hz (A), 10 Hz (B), and 20 Hz (C). Bottom trace showing the GC response to sinusoidal current injection paired with light stimulation in the presence of 100 μM picrotoxin (PTX). (right) Summary graph of spike probability against stimulus number. (D–F) Summary plots of spike probability at initial and last three stimulus numbers at 5 Hz (D), 10 Hz (E), and 20 Hz (F). The spike probabilities of stimulus numbers 1–3 and 8–10 were averaged. (D) one-way ANOVA, n=14, $F_{(2,123)}$ = 28.8, p<0.001 (#1–3), $F_{(2,123)}$ = 25.8, p<0.001 (#8–10); Tukey's post hoc test, ***p<0. 001, **p<0. 01. (E) one-way ANOVA, n=13, $F_{(2,114)}$ = 26.3, p<0.001 (#1–3), $F_{(2,114)}$ = 11.8, p<0.001 (#8–10); Tukey's post hoc test, ***p<0.001, **p<0.01, *p<0.05, n.s., not significant. (F) one-way ANOVA, n=11, $F_{(2,96)}$ = 31.4, p<0.001 (#1–3), $F_{(2,96)}$ = 9.2, p<0.001 (#8–10); Tukey's post hoc test, ***p<0. 001, **p<0.01, n.s., not significant. Data are presented as mean ± SEM.

The online version of this article includes the following source data for figure 6:

**Source data 1.** Source data displayed on *Figure 6*.

later cycle, even during the short-term depression of glutamate release. Thus, the excitatory action of SuM inputs was filtered by the co-released GABA during repetitive high-frequency stimulation due to frequency-independent sustained depression. These results indicate that SuM inputs modulate GC output in a frequency-dependent manner due to the different short-term plasticities of glutamate/GABA co-transmission.

## Discussion

Despite evidence that glutamate and GABA are co-released from the synaptic terminals of a subset of neurons in the adult brain, whether glutamate and GABA are released from the same or distinct

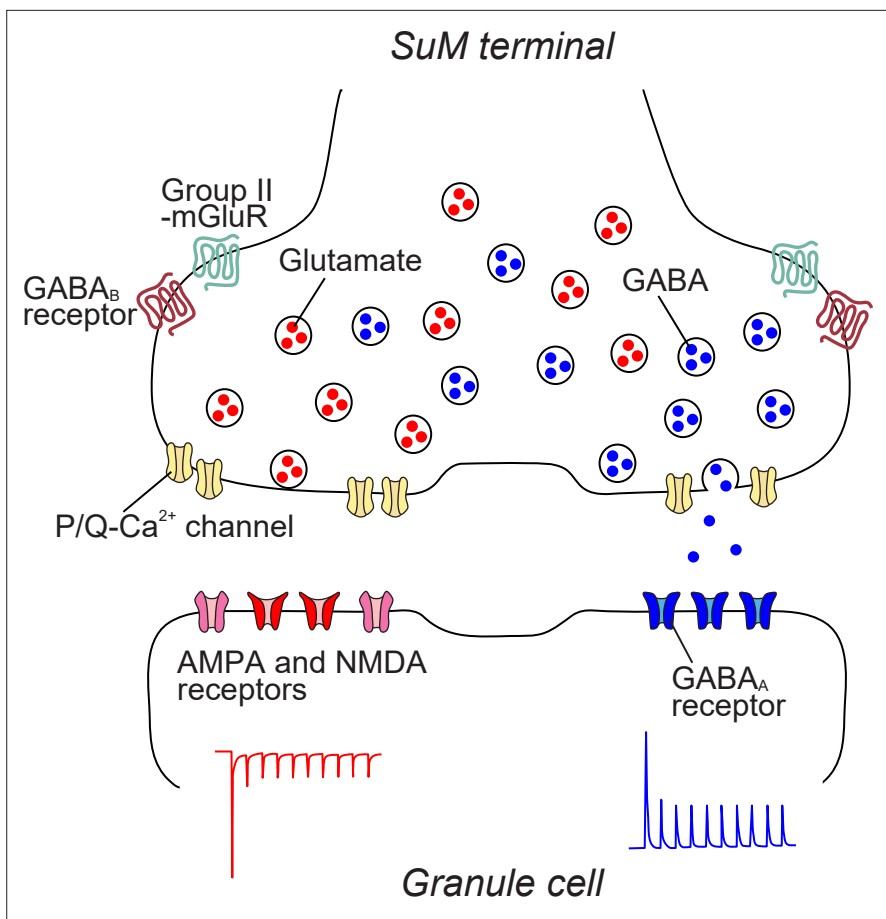

**Figure 7.** Working hypothesis of synaptic architecture of SuM-GC synapse. A single SuM terminal contains distinct glutamatergic and GABAergic vesicles, which are regulated by P/Q-type $Ca^{2+}$ channels and also modulated by group II mGluRs and $GABA_B$ receptors. Glutamatergic and GABAergic vesicles are loosely and tightly coupled with $Ca^{2+}$ channels, respectively. At the postsynaptic site of GCs, AMPA/NMDA and $GABA_A$ receptors are distributed separately. Based on their molecular composition, glutamatergic and GABAergic synapses are established independently, achieving the distinct glutamatergic and GABAergic transmission.

synaptic vesicle populations remains a debate (*Shabel et al., 2014*; *Root et al., 2018*; *Kim et al., 2022*). In this study, we demonstrate that the transmission properties of glutamate and GABA at SuM-GC synapses differ in PPR, $Ca^{2+}$-sensitivity, presynaptic GPCR regulation, and $Ca^{2+}$ channel-synaptic vesicle coupling configuration. We further found that uniquantal synaptic responses exhibit independent EPSCs and IPSCs. The anatomical results show that a single SuM terminal forms both asymmetric and symmetric synapses with a GC soma. The postsynaptic excitatory site (marked by GluN1) and inhibitory site (marked by $GABA_A\alpha1$), facing the same SuM terminal, are closely positioned but distinct. These findings provide evidence that glutamatergic and GABAergic co-transmission occurs at functionally distinct release sites (*Figure 7*). Owing to the functional and spatial segregation of the co-transmission of glutamate and GABA, each transmission exhibits distinct short-term plasticities, leading to different frequency-dependent effects of SuM inputs on the GC activity.

Different release properties of dual-neurotransmitter neurons have been reported in other synaptic circuits. Different PPRs (*Lee et al., 2010*; *Silm et al., 2019*; *Zych and Ford, 2022*) and $Ca^{2+}$-sensitivities (*Lee et al., 2010*; *Zych and Ford, 2022*) between co-transmission of dual-neurotransmitters are found in the retina (acetylcholine and GABA) and striatum (dopamine and glutamate or GABA). Different reliance on $Ca^{2+}$ channels for distinct neurotransmitters have been reported in the retina (acetylcholine and GABA; *Lee et al., 2010*), striatum (dopamine and glutamate; *Silm et al., 2019*), and hippocampus (acetylcholine and GABA; *Takács et al., 2018*). These differential release properties of the two neurotransmitters indicate a separate release from distinct

vesicle pools (*Lee et al., 2010*; *Takács et al., 2018*; *Silm et al., 2019*). Our study also demonstrated that glutamate and GABA co-release from SuM terminals shows differences in PPR and $Ca^{2+}$-sensitivity. Glutamatergic transmission exhibited a stronger depression than GABAergic transmission, suggesting that release probability of glutamate is higher than that of GABA. A higher reduction in IPSCs than EPSCs by decreasing the extracellular $Ca^{2+}$ concentration from 2.5 to 1 mM indicates that GABA release requires higher extracellular $Ca^{2+}$ than glutamate release. Furthermore, using the $Ca^{2+}$ chelators BAPTA and EGTA, we found that the spatial coupling between $Ca^{2+}$ channels and synaptic vesicles of GABA is tighter than that of glutamate. These differences in release properties can be attributed to the different compositions of the release machinery and different spatial organizations between the release machinery and $Ca^{2+}$ channels for glutamate and GABA release (*Südhof, 2013*; *Neher and Brose, 2018*). Given that GABAergic co-transmission exhibits a lower release probability and a higher requirement for extracellular $Ca^{2+}$, together with tight coupling between $Ca^{2+}$ channels and vesicles, synaptic vesicles for GABA release could be surrounded by a few $Ca^{2+}$ channels in close proximity. In contrast, the synaptic vesicles for glutamate release could be surrounded by a large number of $Ca^{2+}$ channels at longer distances than those for GABA release. As with other possible mechanisms, the different expression of transporters in the vesicles with their interacting proteins may contribute to different release probabilities. Given that VGluT1 and endophilin interaction exhibits different release probabilities compared to VGluT2-expressing synapses (*Weston et al., 2011*), it is postulated that different interactions of VGluT2 and VIAAT with their different binding proteins could determine distinct release probabilities.

How are VGluT2 and VIAAT sorted into the distinct synaptic vesicles, and how are glutamate vesicles and GABA vesicles sorted to different places within the same SuM terminal? With synaptic vesicles containing 410 different proteins (*Takamori et al., 2006*), even slight differences in molecular composition or incorporation of specific molecules on the vesicles could influence their functional properties (*Chamberland and Tóth, 2016*). These differences may determine the selective sorting of neurotransmitter transporters into vesicles and the precise arrangement of glutamate and GABA vesicles at the distinct active zones within the same terminal. Especially, VGluT2 and VIAAT contain sorting motifs in their C-terminus, which interacts with clathrin adaptor proteins (APs). Several studies demonstrated that the interaction between neurotransmitter transporters and APs is important for endocytosis and recycling of synaptic vesicles (*Nakatsu et al., 2004*; *Voglmaier et al., 2006*; *Santos et al., 2013*; *Li et al., 2017*; *Silm et al., 2019*). Distinct APs-dependent recycling pathway may contribute to formation of precise spatial localization of glutamate and GABA vesicles in the same SuM terminal (*Asmerian et al., 2024*). Our immuno-EM analysis showed that VGluT2 and VIAAT appear to be randomly distributed in the terminal (*Figure 4C and D*). However, it is important to acknowledge that due to low labeling efficiency, our immuno-EM images may not capture the full spectrum of synaptic vesicles for glutamate and GABA. It is known that synaptic vesicles are divided into three categories: the readily releasable pool (RRP), recycling pool, and resting pool (*Alabi and Tsien, 2012*; *Miki et al., 2022*). Among them, the vast majority of synaptic vesicles (~85% of the total vesicles) belongs to the resting pool. Therefore, it is likely that most of the VGluT2 and VIAAT labeling corresponds to the synaptic vesicles in the resting pool. Further studies are necessary to investigate whether VGluT2-containing vesicles and VIAAT-containing vesicles in the RRP and recycling pools, both of which are closely linked to exocytosis, are clearly segregated near their respective active zones.

Our data showed that both glutamate and GABA release rely on P/Q-type $Ca^{2+}$ channels at SuM-GC synapses. Interestingly, the excitatory transmission of SuM-CA2 pyramidal neuron synapses, an exclusive glutamatergic synapse (*Chen et al., 2020*), is also mediated by P/Q-type $Ca^{2+}$ channels (*Robert et al., 2021*). Therefore, neurotransmitter release from SuM neurons depends on P/Q-type $Ca^{2+}$ channels, irrespective of the target neurons. We found that the activation of group II mGluRs and $GABA_B$ receptors inhibited glutamatergic co-transmission more strongly than GABAergic co-transmission. Given that these GPCRs suppress presynaptic $Ca^{2+}$ channels via $G_{i/o}$-protein, and that both transmitter releases are mediated by the same type of $Ca^{2+}$ channels, the different extents of inhibition of the co-release of glutamate/GABA by group II mGluRs and $GABA_B$ receptors may be attributed to differential expression levels of receptors or signaling efficacy between glutamate and GABA release. Additionally, the spatial coupling between GPCRs and active zones for glutamate and GABA in the same SuM terminals may be different, which may give rise to differential modulation of glutamate and GABA release.

Consistent with previous EM studies (*Boulland et al., 2009*; *Soussi et al., 2010*; *Root et al., 2018*; *Billwiller et al., 2020*), we found that a single SuM terminal forms symmetric and asymmetric synapses in the DG (*Figure 4E and F*). However, confirming whether these synapses function as glutamatergic or GABAergic relies on the identification of their postsynaptic receptors, which remains unresolved. In our study, we demonstrated that both GluN1 and GABA$_A$α1 were associated with common SuM terminals (*Figure 4G and H*), suggesting that these synapses represent distinct glutamatergic and GABAergic synapses. Furthermore, our analysis showed that approximately 50% of SuM terminals was associated with both GluN1 and GABA$_A$α1 (*Figure 4I*). It should be noted, however, this percentage can be an underestimate due to the utilization of 100-nm-thick ultrathin sections, which limits the detection of co-localization events within this confined spatial range. Consequently, the actual percentage is expected to be significantly higher. Hence, the predominant synapse type of SuM terminals consists of dual glutamatergic and GABAergic synapses, with additional individual glutamatergic or GABAergic synapses.

Previous electrophysiological studies have demonstrated that both glutamate and GABA are co-packaged in the same synaptic vesicles in the lateral habenula (*Shabel et al., 2014*; *Kim et al., 2022*). Minimal optogenetic stimulation of EP terminals evoked a high proportion of biphasic PSCs over independent EPSCs and IPSCs in lateral habenular neurons, suggesting the co-release of glutamate and GABA from the same synaptic vesicles (*Kim et al., 2022*). While these findings appear to contradict anatomical data showing distinct populations of glutamate- and GABA-containing synaptic vesicles in the lateral habenula (*Root et al., 2018*), technical limitations of immuno-EM may affect the ability to accurately assess the spatial distributions of proteins within the vesicles (*Kim and Sabatini, 2022*). Such limitations include antibody interference on small vesicles, restricted access to target molecules, and the stochastic nature of labeling two molecules in a single vesicle. In addition, the existence of VTA terminals, other glutamate/GABA co-release inputs (*Root et al., 2014*; *Yoo et al., 2016*; *Root et al., 2018*), may complicate the interpretation of the co-release mechanisms in the lateral habenula. In contrast, in the DG, we found that the majority of synaptic responses evoked by both minimal light stimulation and asynchronous release at SuM-GC synapses showed independent EPSCs and IPSCs. A previous biochemical study detected VGluT2 and VIAAT in distinct purified synaptic vesicles from the hippocampus (*Boulland et al., 2009*). Moreover, differences in transport mechanisms between glutamatergic and GABAergic vesicles have been reported (*Farsi et al., 2016*). Based on this evidence, our results most likely support the hypothesis that glutamate and GABA are packaged in distinct synaptic vesicles (*Figure 7*). Thus, glutamate/GABA co-transmission is diverse, in that both transmitters are co-released from the same or different synaptic vesicles, depending on brain regions and synapse types. It is important to note that transmitter release is detected by postsynaptic receptors; therefore, even if two neurotransmitters are co-released from the same vesicles, they will not be detected unless the postsynaptic targets express receptors for both (*Dugué et al., 2005*; *Granger et al., 2020*). Accordingly, while our results strongly suggest that glutamate/GABA co-transmission occurs independently at segregated excitatory and inhibitory SuM-GC synapses, it remains possible that glutamate and GABA are co-released from the same vesicles, with one transmitter undetected due to the absence of its postsynaptic receptor. Further study will be necessary to determine the packaging mechanisms of glutamate and GABA within SuM terminals.

The distinct co-transmission modes of the two transmitters have the ability to exert independent synaptic regulation compared with their co-release from identical vesicles. At SuM-GC synapses, we demonstrated that glutamatergic and GABAergic co-transmission exhibited different short-term plasticities. While glutamatergic co-transmission showed strong frequency-dependent depression, GABAergic co-transmission exhibited stable frequency-independent depression. By this frequency-specific dynamic change of the glutamate/GABA co-transmission balance, at low-frequency burst SuM inputs, they operated as excitatory inputs, whereas at high frequency, the excitatory effects of SuM inputs on GCs were inhibited by co-transmitted GABA (*Figure 6*). Accordingly, GCs respond differently to SuM inputs in a frequency-dependent manner, due to the low-pass filtering of SuM-GC synapses. The fact that the theta-rhythm in the hippocampus is associated with synchronized SuM activity (*Kirk and McNaughton, 1991*; *Kocsis and Vertes, 1994*; *Ito et al., 2018*; *Farrell et al., 2021*) has important implications for the low-pass filtering of this synapse. As SuM inputs act as excitatory at 5 Hz (theta) train stimulation, but not at higher frequencies, SuM inputs may transfer their train information exclusively at the theta frequency, which could contribute to theta-mediated information

processing in SuM-hippocampal circuits and their relevant brain functions, including cognitive, sleep, and navigation processes (*Pedersen et al., 2017*; *Billwiller et al., 2020*; *Chen et al., 2020*; *Li et al., 2020*; *Farrell et al., 2021*; *Kesner et al., 2023*). In addition to SuM inputs, GCs receive EC inputs as the main excitatory inputs for discharge, and other excitatory inputs from hilar mossy cells (*Hashimotodani et al., 2017*). Therefore, low-pass filtering of SuM inputs could prevent the over-excitation of GCs, leading to anti-seizure effects. In contrast to the presynaptic origin of short-term plasticity at SuM-GC synapses, we previously demonstrated the postsynaptically induced long-term potentiation at SuM-GC synapses, which selectively occurs of glutamatergic co-transmission (*Hirai et al., 2022*; *Tabuchi et al., 2022*). Our findings suggest that SuM inputs play diverse roles in modulating GC output via short- and long-term plasticity with pre- and postsynaptic distinct mechanisms, respectively.

## Materials and methods

### Animals

C57BL/6 and VGluT2-Cre mice (Jackson Labs, Slc17a6$^{tm2(cre)Lowl}$/J, stock #016963) of either sex aged 6–7 weeks were used for electrophysiological experiments and male C57BL/6 mice at 2–3 months of age were used for anatomical experiments (*Figure 4*). All animals were group housed in a temperature- and humidity-controlled room under a 12 hr light/12 hr dark cycle. Water and food were provided ad libitum. The experiments were approved by the animal care and use committee of Doshisha University and Hokkaido University and were performed in accordance with the guidelines of the committees.

### Stereotaxic injections

Mice on postnatal days 19–21 were placed in a stereotaxic frame, and anesthetized with isoflurane (1.5–2.5%). A beveled glass capillary pipette connected to a microsyringe pump (UMP3, WPI) was used for the viral injection. 200 nL of AAV1-EF1a-DIO-hChR2(H134R)-eYFP (Addgene) was injected into the SuM (relative to bregma, AP: −2.2 mm, ML: ±0.5 mm, DV: −4.85 mm) at a rate of 50 nL/min. The glass capillary was remained at the target site for 5 min before the beginning of the injection and was removed 10 min after infusion. For retrograde tracing, 900 nL of 0.8% fluorescent microspheres (Fluospheres, 0.04 µm, 565/580, Invitrogen, F8794) was injected unilaterally into the dorsal DG (relative to bregma, AP: −1.96 mm, ML: 1.484 mm, DV: 1.93 mm) at a rate of 100 nL/min using a 10 µl Hamilton microsyringe with a beveled 32 gauge needle. Animals were sacrificed two weeks later for fluorescent retrograde tracing.

### Hippocampal slice preparation

Acute transverse hippocampal slices (300-µm-thick) were prepared from mice 3–4 weeks after AAV injection. The mice were decapitated under isoflurane anesthesia. Briefly, the hippocampi were isolated, embedded in an agar block, and cut using a vibratome (VT1200S, Leica Microsystems) in an ice-cold cutting solution containing (in mM): 215 sucrose, 20 D-glucose, 2.5 KCl, 26 NaHCO$_3$, 1.6 NaH$_2$PO$_4$, 1 CaCl$_2$, 4 MgCl$_2$, and 4 MgSO$_4$. Brain blocks, including the interbrain and midbrain, were also isolated and fixed in 4% paraformaldehyde (PFA) for *post hoc* our internal check of the injection site (*Tabuchi et al., 2022*). Hippocampal slices were transferred to an incubation chamber and incubated at 33.5 °C in the cutting solution. After 30 min of incubation, the cutting solution was replaced with an extracellular artificial cerebrospinal fluid (ACSF) containing (in mM): 124 NaCl, 2.5 KCl, 26 NaHCO$_3$, 1 NaH$_2$PO$_4$, 2.5 CaCl$_2$, 1.3 MgSO$_4$ and 10 D-glucose at 33.5 °C. The slices were stored at room temperature for at least 1 hr before recording. Both the cutting solution and the ACSF were oxygenated with 95% O$_2$ and 5% CO$_2$. After recovery, slices were transferred to a submersion-type recording chamber for electrophysiological analysis.

### Electrophysiology

Whole-cell recordings using an EPC10 (HEKA Electronik) or IPA amplifier (Sutter Instruments) were made from GCs using an infrared differential interference contrast microscopy (IR-DIC, Olympus, BX51WI). We recorded from GCs with an input resistance of <300 MΩ for mature GCs (*Schmidt-Hieber et al., 2004*). The data were filtered at 2.9 kHz and sampled at 20 kHz. For voltage-clamp recordings, we used patch pipettes (3–6 MΩ) filled with an intracellular solution with the following composition (in mM): 131 Cs-gluconate, 4 CsCl, 10 HEPES, 0.2 EGTA, 2 Mg-ATP, 0.3 Na$_3$GTP, 10

phosphocreatine, pH 7.3 adjusted with CsOH (the calculated $E_{Cl^-}$ = −90 mV; 292–300 mOsm). In some experiments, biocytin (0.5%) was included in the intracellular solution for morphological visualization of GCs after recording using Alexa Fluor 568-conjugated streptavidin (1:500, Thermo Fisher Scientific). For recordings of synaptic currents at intermediate membrane potentials, 5 mM QX314 was included in the intracellular solution. ChR2-expresing SuM fibers were activated at 0.05 Hz by a pulse of 470 nm blue light (1–5ms duration, 5.0–10.5 mW/mm$^2$) delivered through a 40× objective attached to a microscope using an LED (Mightex or ThorLabs). The light intensity was adjusted to yield less than 80% of the maximum response during the baseline period. The illumination field was centered over the recorded cell, unless otherwise stated. Series resistance (8–20 MΩ) was uncompensated and monitored throughout experiments with a −5 mV, 50ms voltage step, and cells that exhibited a significant change in the series resistance more than 20% were excluded from analysis.

For minimal light stimulation experiments, we first recorded biphasic synaptic responses from GCs by maximum light stimulation of SuM inputs at intermediate membrane potentials from −20 to −30 mV in the presence of 1 μM TTX, 1 mM 4-AP, 50 μM D-AP5, and 1 mM Ca$^{2+}$. The exact membrane potentials were set to the potential at which the amplitudes of EPSCs and IPSCs were similar. Then, light power intensity was decreased to detect small stochastic synaptic events. We adjusted the light power (<1 ms duration, <1.0 mW/mm$^2$) to evoke synaptic responses, with a success rate of 15%. A total of 100 trials were recorded every 10 s. EPSC-only responses were determined as the synaptic current with the onset of a downward deflection within 6 ms of light illumination that returned to the baseline without an outward current within 20 ms of light illumination. IPSC-only responses were determined as the synaptic current with the onset of an upward deflection within 6 ms of light illumination, without an inward current immediately after light illumination. Biphasic responses were defined as an EPSC-IPSC sequence, only if an outward peak current following an inward current appeared within 20 ms of light illumination. The amplitude threshold was set to three times the SD of the baseline noise. Based on these criteria, we confirmed that the detection rate of false-positive events (without light illumination) was 0.6 ± 0.4% (3/500 trials, n=5).

For recordings of asynchronous synaptic events, we initially recorded light stimulation-evoked biphasic synaptic responses at intermediate membrane potentials (−20 to −30 mV) in the presence of Ca$^{2+}$. The ACSF was then replaced with Ca$^{2+}$-free ACSF containing 4 mM SrCl$_2$. Asynchronous synaptic responses were evoked by light pulses at 0.1 Hz for 30 times, measured during a 450 ms period beginning 30 ms after the stimulus to exclude the initial synchronous synaptic responses, and analyzed using the Mini Analysis Program (Synaptosoft). D-AP5 (50 μM) was included in the extracellular ACSF throughout the experiment to eliminate NMDA receptor-mediated currents.

For train stimulation, EPSCs were recorded from GCs at a holding potential of −70 mV in the presence of 100 μM picrotoxin, while IPSCs were recorded at a holding potential of 0 mV in the presence of 10 μM NBQX and 50 μM D-AP5. Biphasic PSCs were recorded at a holding potential of −30 mV in the presence of 1 μM TTX, 500 μM 4-AP, and 50 μM D-AP5. In all train-stimulation experiments, 100 μM LY341495 and 3 μM CGP55845 were also added to the ACSF throughout the experiments. 10 light pulses at 5 Hz, 10 Hz, and 20 Hz were delivered every 20 s, and 10–20 sweeps were recorded.

For field potential recordings in the CA1 region, a recording pipette filled with 1 M NaCl and a patch pipette with broken tip (with diameter of ~20–30 μm) filled with the ACSF were placed in the stratum radiatum of the CA1 region. For recordings of the fiber volley originating from ChR2-epressing SuM fibers, a recording pipette was placed in the supragranular layer where ChR2-epressing SuM fibers were densely distributed. Light-evoked field potentials were recorded in the presence of 10 μM NBQX, 50 μM D-AP5, and 100 μM picrotoxin to block the synaptic responses.

For current-clamp recordings, we used an intracellular solution with the following composition (in mM): 135 KMeSO$_4$, 5 KCl, 10 HEPES, 0.2 EGTA, 2 MgATP, 0.3 Na$_3$GTP, 10 phosphocreatine, pH 7.3 adjusted with NaOH (the calculated $E_{Cl^-}$ = −82 mV; 297–299 mOsm). A sinusoidal current was delivered to the soma of GCs at 5 Hz, 10 Hz, and 20 Hz (10 cycles), while the membrane potential was held at −70 mV to −80 mV. The injected current (100–240 pA) was adjusted to generate action potentials with fewer than five spikes in each trial. After obtaining control, a sinusoidal current injection was paired with light stimulation of SuM inputs at the depolarizing phase of each cycle. Subsequently, 100 μM picrotoxin was bath applied, and the same pairing of sinusoidal current with light stimulation were delivered. In each experiment, 10 trials were recorded every 20 s.

**Table 1.** List of primary antibodies used in the present study.

| Molecules | Sequence (NCBI #) | Host | RRID | Specificity | Source | Figure |
|---|---|---|---|---|---|---|
| GABA$_A$α1 | | Gp | AB_2571572 | IB | *Ichikawa et al., 2011* | *Figure 4* |
| GluN1 | | Ms | AB_2571605 | | Millipore (MAB363) | *Figure 4* |
| VGluT2 | 559–582 aa (BC038375) | Gp | AB_2571621 | IB | *Miyazaki et al., 2003*; (MSFR106280) | *Figure 4* |
| | | Go | AB_2571620 | IB | *Kawamura et al., 2006*; (MSFR106270) | *Figure 4* |
| | | Rb | AB_2571619 | IB | *Miyazaki et al., 2003*; (MSFR106300) | *Figure 1* |
| VGAT/VIAAT | 31–112 aa (BC052020) | Rb | AB_2571622 | IB | *Miura et al., 2006*; (MSFR106120) | *Figure 4* |
| | | Gp | AB_2571624 | IB | *Miura et al., 2006*; (MSFR106150) | *Figure 1, Figure 4* |
| GFP | | Go | AB_305643 | | Abcam (ab6673) | *Figure 1* |
| | | Rb | AB_2491093 | | *Takasaki et al., 2010* | *Figure 4* |

All experiments were performed at 30–33 °C in the recording chamber perfused (2 mL/min) with oxygenated ACSF.

## Pharmacology

Each reagent was dissolved in water, NaOH or DMSO, depending on the manufacture's recommendation to prepare a stock solution, and stored at −20 °C. NBQX, D-AP5, (R)-baclofen, DCG-IV, γDGG, TPMPA, and LY341495 were purchased from Tocris Bioscience. $\omega$-Conotoxin GVIA and $\omega$-agatoxin IVA were purchased from the Peptide Institute. Picrotoxin was purchased from the Tokyo Chemical Industry. 4-AP and BAPTA-AM were purchased from nacalai tesque. TTX was purchased from Fujifilm Wako Pure Chemical. EGTA-AM was purchased from AAT Bioquest. CGP55845 was purchased from Hello Bio. Reagents were bath applied following dilution into ACSF from the stock solutions immediately before use.

## Antibodies

Primary antibodies raised against the following molecules were used: GABA$_A$ receptor α1 subunit, NMDA receptor GluN1 subunit, VGluT2, VIAAT, and GFP. Information on the molecule, antigen sequence, host species, specificity, reference, NCBI GenBank accession number, and RRID of the primary antibodies used is summarized in *Table 1*.

## Immunohistochemistry for ChR2-expressing SuM terminals

Under deep pentobarbital anesthesia (100 mg/kg of body weight, intraperitoneally), AAV-DIO-ChR2(H134R)-eYFP injected VGluT2-Cre mice were perfused with 4% PFA. Brains were removed and stored in 4% PFA for 4 hr at room temperature, then transferred to PBS and left overnight at 4 °C. Coronal brain slices containing the SuM or the hippocampus were sectioned at 100 µm using a vibratome (VT1200S, Leica microsystems). After permeabilization in PBS with 1% Triton X-100 for 30 min, the sections were blocked in PBS with 5% donkey serum for 1 h. After washing three times with PBS for 10 min each, the sections were incubated in PBS containing primary antibodies and 0.5% Triton X-100 overnight at room temperature. Primary antibodies against GFP (goat, 1:200), VGluT2 (rabbit, 1:200), and VIAAT (guinea pig, 1:200) were used. The sections were then rinsed with PBS three times, and incubated in PBS containing 0.5% Triton X-100 and secondary donkey antibodies (1:500, anti-goat Alexa Fluor 488-conjugated antibody, Abcam, ab150129; anti-rabbit Alexa Fluor 568-conjugated antibody, Abcam, ab175470; anti-guinea pig Cy5-conjugated antibody, Jackson ImmunoResearch, 706-175-148) for 2 hr at room temperature. Sections were rinsed three times in PBS and mounted on glass slides with DAPI. To analyze colocalization of ChR2-eYFP-labeled axonal boutons with VGluT2 and VIAAT, z stack images were captured (0.4 µm optical sections) using a fluorescence microscope (BZ-X800, Keyence). Swellings of ChR2-YFP labeled axons were considered as

axonal boutons, and their colocalization with VGluT2 and VIAAT immunofluorescence was examined in 3D planes as previously reported (*Hashimotodani et al., 2018*).

## Immunofluorescence using ultrathin sections

Under deep pentobarbital anesthesia (100 mg/kg body weight, i.p.), mice were fixed by transcardially with 5 ml of saline, followed by 60 ml of glyoxal fixative solution (9% glyoxal [Sigma], 8% acetic acid, pH 4.0; *Konno et al., 2023*). Brains were dissected and post-fixed at 4 °C in glyoxal fixative solution for 2 hr. Coronal 400-μm-thick brain sections were cut on a vibratome (VT1000S, LeicaMicrosystems). Sections were embedded in durcupan (Sigma) and ultra-thin sections (100-nm-thick) were prepared by Ultracut ultramicrotome (Leica Microsystems). Sections were mounted on silane-coated glass slides (New Silane II, *Muto* Pure Chemicals). After etching with saturated sodium ethanolate solution for 1–5 s, ultra-thin sections on slides were treated with ImmunoSaver (Nisshin EM) at 95 °C for 30 min. Sections were blocked with 10% normal donkey serum (Jackson ImmunoResearch) for 20 min, then incubated with a mixture of primary antibody solution (1 μg/ml in phosphate-buffered saline (pH 7.2)) containing 0.1% TritonX-100 (PBST) overnight at room temperature. Sections were washed three times with PBST and incubated with a solution of Alexa488, Cy3, and Alexa647-labeled species-specific secondary antibodies (Jackson ImmunoResearch; Thermo Fisher Scientific) diluted at 1:200 in PBST for 2 hr at room temperature. After washing, sections were air-dried and mounted using ProLong Glass (Thermo Fisher Scientific). Photographs were taken with a confocal laser microscope (FV1200, Evident) using 20x and 60x objectives.

## Conventional electron microscopy

Under deep anesthesia with pentobarbital (100 mg/kg, i.p.), mice were transcardially perfused with 2% PFA/2% glutaraldehyde in 0.1 M PB (pH 7.2) for 10 min, and the brains were removed. Brains were then postfixed for 2 hr in the same fixative, and coronal sections (50-μm-thick) were prepared using a vibratome (VT1000S, Leica Microsystems). Sections were fixed with 1% osmium tetroxide solution for 15 min, and then stained with 2% uranyl acetate for 15 min, dehydrated in a graded ethanol series and n-butyl glycidyl ether, embedded in epoxy resin, and polymerized at 60 °C for 48 hr. Ultrathin sections (~80-nm-thick) were prepared with an ultramicrotome (UCT7, Leica Microsystems). Serial sections were mounted on indium-tin-oxide-coated glass slides (IT5-111-50, NANOCS) and successively stained with 2% uranyl acetate and lead citrate. After washing, colloidal graphite (Ted Pella Inc) was pasted on the glass slides to surround the ribbons. Images were acquired using a SEM with a backscattered electron beam detector at an accelerating voltage of 1.0 kV and a magnification of ×12,000 (SU8240, Hitachi High Technologies).

## Postembedding immunoelectron microscopy

Under deep anesthesia with pentobarbital (100 mg/kg, i.p.), mice were transcardially perfused with 4% PFA/0.1 M PB (pH 7.2) for 10 min, brains were removed, postfixed for 2 hr in the same fixative, and coronal sections (400-μm-thick) prepared using a vibratome (VT1000S, Leica Microsystems). Then, sections were cryoprotected with 30% glycerol in PB, and frozen rapidly with liquid propane in the EM CPC unit (Leica Microsystems). Frozen sections were immersed in 0.5% uranyl acetate in methanol at –90 °C in the AFS freeze-substitution unit (Leica Microsystems), infiltrated at –45 °C with Lowicryl HM-20 resin (Chemische Werke Lowi, Waldkraiburg, Germany), and polymerized with ultraviolet light. Ultrathin sections were made using an Ultracut ultramicrotome (Leica Microsystems). Ultrathin sections (100-nm-thick) mounted on nickel grids were etched with ultrathin sections on nickel grids were etched with saturated sodium-ethanolate solution for 1–5 s. For double labeling against VGluT2 and VIAAT, sections were incubated in blocking solution containing 2% normal goat serum (Nichirei, Tokyo, Japan) in 0.03% Triton X-100 in Tris-buffered saline (TBST, pH 7.4), followed by VIAAT antibody (20 μg/ml) diluted in 2% normal goat serum in TBST overnight, then colloidal gold-conjugated (5 nm) anti-rabbit IgG in blocking solution for 2 hr. After extensive washing in distilled water, sections were incubated in blocking solution containing 2% rabbit serum (Nichirei, Tokyo, Japan) in TBST for 10 min, VGluT2 antibody (20 μg/ml) diluted with 2% normal rabbit serum in TBST overnight, and colloidal gold-conjugated (10 nm) anti-guinea pig IgG in blocking solution for 2 hr. After extensive washing in distilled water, sections were fixed with 2% OsO4 for 15 min, then stained with 5% uranyl acetate/40%

EtOH for 90 s and Reynold's lead citrate solution for 60 s. Photographs were taken from the granular layer of the dentate gyrus with an H-7100 electron microscope at x20,000 magnification.

## Pre-embedding immunoelectron microscopy

Under deep anesthesia with pentobarbital (100 mg/kg, i.p.), VGluT2-Cre mice, in which AAV1-EF1a-DIO-hChR2(H134R)-eYFP was injected into the SuM were transcardially perfused with 4% PFA/0.1 M PB (pH 7.2) for 10 min, and the brains were removed. Brains were then postfixed for 2 hr in the same fixative, and coronal sections (50-μm-thick) were prepared using a vibratome (VT1000S, Leica Microsystems). PBS (pH 7.4) containing 0.1% Tween 20 was used as incubation and washing buffer. Sections were blocked with 10% normal goat serum (Nichirei Bioscience Corporation) for 20 min and incubated with rabbit anti-GFP antibody (1 μg/ml) overnight at room temperature. After washing, sections were incubated with colloidal-conjugated anti-rabbit IgG (1:100; Nanoprobes) for 2 hr at room temperature. Sections were washed with HEPES Buffer (50 mM HEPES, 200 mM sucrose, 5 N sodium hydroxide; pH 8.0), and then incubated with silver enhancement reagent (AURION R-Gent SE-EM; AURION) for 1 hr. Sections were fixed with 1% osmium tetroxide solution on ice for 15 min, and then stained with 2% uranyl acetate for 15 min, dehydrated in a graded ethanol series and n-butyl glycidyl ether, embedded in epoxy resin, and polymerized at 60 °C for 48 hr. Ultrathin sections (~80-nm-thick) in the plane parallel to the section surface were prepared with an ultramicrotome (UCT7, Leica Microsystems). Serial sections were mounted on indium-tin-oxide-coated glass slides (IT5-111-50, NANOCS) and successively stained with 2% uranyl acetate and lead citrate. After washing, colloidal graphite (Ted Pella Inc) was pasted on the glass slides to surround the serial sections. Images were acquired using a SEM with a backscattered electron beam detector at an accelerating voltage of 1.0 kV and a magnification of ×12,000 (SU8240, Hitachi High Technologies).

## Data analysis

The PPR was defined as the ratio of the amplitude of the second PSCs to the amplitude of the first PSCs (100 ms inter-stimulus interval). The reduction in PSCs by a decrease in extracellular $Ca^{2+}$ concentration was determined by comparing 10 min baseline responses (2.5 mM $Ca^{2+}$) with responses 20–25 min after replacement of extracellular $Ca^{2+}$ (1 mM). The inhibition of PSCs by DCG-IV, baclofen, $\omega$-CgTx, $\omega$-Aga-IVA, BAPTA-AM, and EGTA-AM was determined by comparing the 10 min baseline responses with responses at 15–20 min (or 25–30 min for BAPTA-AM, and EGTA-AM) after each drug application. The PSCs probability was calculated as the percentage of the number of synaptic responses among 100 sweeps. Train stimulation-induced depression was calculated as the ratio of the amplitude of the second to 10th PSCs to the amplitude of the first PSCs. Each stimulus point had an averaged of 10–20 sweeps. Spike probability was calculated as the percentage of the number of spikes among 10 trials.

## Statistics

Statistical analyses were performed using OriginPro software (OriginLab, USA). Normality of the distributions was assessed using the Shapiro−Wilk test. For samples with normal distributions, Student's unpaired and paired two-tailed t tests were used to assess the between-group and within-group differences, respectively. For samples that were not normally distributed, a non-parametric paired sample Wilcoxon signed-rank test was used. The Kolmogorov−Smirnov test was used for cumulative distributions. Differences among two or multiple samples were assessed by using one-way or two-way ANOVA followed by Tukey's *post hoc* test for multiple comparisons. Statistical significance was set at $p < 0.05$ (***, **, and * indicates $p < 0.001$, $p < 0.01$ and $p < 0.05$, respectively). All values are reported as the mean ± SEM.

## Acknowledgements

This work was supported by the Grants-in-Aid for Scientific Research (JP24KJ2139 to HH; JP20KK0171 and JP24K02125 to TS; JP20H03358, JP23H04240, and JP23K18167 to YH) from Japan Society for the Promotion of Science (JSPS), the Takeda Science Foundation (to YH and TS), the Naito Foundation (to YH), the Mochida Memorial Foundation for Medical and Pharmaceutical Research (to YH), JSPS Core-to-Core Program A. Advanced Research Networks (grant number: JPJSCCA20220007 to

TS) and Grant-in-Aid for Transformative Research Areas—Platforms for Advanced Technologies and Research Resources "Advanced Bioimaging Support" (grant number: JP22H04926).

## Additional information

### Funding

| Funder | Grant reference number | Author |
|---|---|---|
| Japan Society for the Promotion of Science | JP24KJ2139 | Himawari Hirai |
| Japan Society for the Promotion of Science | JP20H03358 | Yuki Hashimotodani |
| Japan Society for the Promotion of Science | JP23H04240 | Yuki Hashimotodani |
| Japan Society for the Promotion of Science | JP23K18167 | Yuki Hashimotodani |
| Japan Society for the Promotion of Science | JP20KK0171 | Takeshi Sakaba |
| Japan Society for the Promotion of Science | JP24K02125 | Takeshi Sakaba |
| Takeda Science Foundation | Medical Research Continuous Grants | Yuki Hashimotodani |
| Takeda Science Foundation | Bioscience and Specific Research Grants | Takeshi Sakaba |
| Naito Foundation | | Yuki Hashimotodani |
| Mochida Memorial Foundation for Medical and Pharmaceutical Research | | Yuki Hashimotodani |
| Japan Society for the Promotion of Science | JSPS Core-to-Core Program A. Advanced Research Networks JPJSCCA20220007 | Takeshi Sakaba |
| Japan Society for the Promotion of Science | Platforms for Advanced Technologies and Research Resources "Advanced Bioimaging Support" JP22H04926 | Masahiko Watanabe |

The funders had no role in study design, data collection and interpretation, or the decision to submit the work for publication.

### Author contributions

Himawari Hirai, Kohtarou Konno, Data curation, Formal analysis, Validation, Investigation, Writing – review and editing; Miwako Yamasaki, Conceptualization, Validation, Investigation, Writing – review and editing; Masahiko Watanabe, Conceptualization, Supervision, Funding acquisition, Writing – review and editing; Takeshi Sakaba, Conceptualization, Funding acquisition, Writing – review and editing; Yuki Hashimotodani, Conceptualization, Data curation, Formal analysis, Supervision, Funding acquisition, Validation, Investigation, Writing – original draft, Writing – review and editing

### Author ORCIDs

Miwako Yamasaki (ID) https://orcid.org/0000-0003-4974-9349
Masahiko Watanabe (ID) https://orcid.org/0000-0001-5037-7138
Takeshi Sakaba (ID) https://orcid.org/0000-0003-0688-7717
Yuki Hashimotodani (ID) https://orcid.org/0000-0001-6723-2736

## Ethics

All animal experiments were performed in accordance with guidelines approved by the animal care and use committee of Doshisha University (A24063, D24063) and Hokkaido University (#19-0111, #23-0033).

Reviewer #1 (Public review): https://doi.org/10.7554/eLife.99711.3.sa1
Reviewer #2 (Public review): https://doi.org/10.7554/eLife.99711.3.sa2
Reviewer #3 (Public review): https://doi.org/10.7554/eLife.99711.3.sa3
Author response https://doi.org/10.7554/eLife.99711.3.sa4

# Additional files

## Supplementary files

• MDAR checklist

## Data availability

All data generated or analysed during this study are included in the manuscript and supporting files.

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
