## [Editor Report · eLife Assessment]

This **fundamental** work provides evidence that glutamate and GABA are released from different synaptic vesicles at supramammillary axon terminals onto granule cells of the dentate gyrus. The study uses complementary electrophysiological and anatomical experimental approaches. Together, these provide **convincing** evidence that the co-release of glutamate and GABA from different vesicles within the same terminal could modulate granule cell firing in a frequency-dependent manner, although thorough elimination of alternative mechanisms would have strengthened the study. The work will be of interest to neuroscientists investigating co-release of neurotransmitters in various synapses in the brain and those interested in subcortical control of hippocampal function.

---

## [Referee Report · Reviewer #1 (Public review)]

This study of mixed glutamate/GABA transmission from axons of the supramammillary nucleus to dentate gyrus seeks to sort out whether the two transmitters are released from the same or different synaptic vesicles. This conundrum has been examined in other dual-transmission cases and even in this particular pathway there are different views. The authors use a variety of electrophysiological and immunohistochemical methods to reach the surprising (to me) conclusion that glutamate and GABA filled vesicles are distinct yet released from the same nerve terminals. While the strength of the conclusion rests on the abundance of data (approaches) rather than the decisiveness of any one approach, I came away believing that the boutons may indeed produce and release distinct types of vesicles. Accepting the conclusion, one is now left with another conundrum: how can a single bouton sort out VGLUTs and VIAATs to different vesicles, position them in distinct locations with nm precision and recycle them without mixing? And why do it this way instead of with single vesicles having mixed chemical content? For example, could a quantitative argument be made that separate vesicles allow for higher transmitter concentrations? Hopefully, future studies will probe these issues.

---

## [Referee Report · Reviewer #2 (Public review)]

Summary:

In this study, the authors investigated the release properties of glutamate/GABA co-transmission at the supramammillary nucleus (SuM)-dentate granule cell (DGC) synapses using state -of-the-arts in vitro electrophysiology and anatomical approaches at the light and electron microscopy level. They found that SuM to dentate granule cell synapses, which co-release glutamate and GABA, exhibit distinct differences in paired-pulse ratio, Ca2+ sensitivity, presynaptic receptor modulation, and Ca2+ channel-vesicle coupling configuration for each neurotransmitter. The study shows that glutamate/GABA co-release produces independent glutamatergic and GABAergic synaptic responses, with postsynaptic targets segregated. They show that most SuM boutons form distinct glutamatergic and GABAergic synapses at proximity, characterized by GluN1 and GABAAα1 receptor labeling respectively. Furthermore, they demonstrate that glutamate/GABA co-transmission exhibits distinct short-term plasticity, with glutamate showing frequency-dependent depression and GABA showing frequency-independent stable depression. The authors provide compelling evidence at the anatomical and physiological levels that glutamate and GABA are co-release by different synaptic vesicles within the same synaptic terminal at the SuM-DGC synapses and that the distinct transmission modes of the glutamate and GABA release serve as a frequency-dependent filters of SuM inputs on GC outputs.

This is a fundamental work, that significantly advances our understanding of the mechanism by which the two fast-acting and functionally opposing neurotransmitters glutamate and GABA are co-transmitted at the SuM-DGC synapses and the functional role of this type of Glutamate/GABA co-transmission.

Strengths:

The conclusions of this paper are provided by a large number of compelling data

---

## [Referee Report · Reviewer #3 (Public review)]

Summary:

In this manuscript, Hirai et al investigated the release properties of glutamate/GABA co-transmission at SuM-GC synapses and reported that glutamate/GABA co-transmission exhibits distinct short-term plasticity with segregated postsynaptic targets. Using optogenetics, whole-cell patch-clamp recordings, and immunohistochemistry, the authors reveal distinct transmission modes of glutamate/GABA co-release as frequency-dependent filters of incoming SuM inputs.

Strengths:

Overall, this study is well-designed and executed; conclusions are supported by the results. This study addressed a long-standing question of whether GABA and glutamate are packaged in the same vesicles and co-released in response to the same stimuli in the SuM-GC synapses (Pedersen et al., 2017; Hashimotodani et al., 2018; Billwiller et al., 2020; Chen et al., 2020; Li et al., 2020; Ajibola et al., 2021). Knowledge gained from this study advances our understanding of neurotransmitter co-release mechanisms and their functional roles in the hippocampal circuits.

Comments on revisions:

The authors have addressed my comments, and now the manuscript is in a good form as it currently stands.

---

## [Author Response]

The following is the authors’ response to the original reviews.

**Public Reviews:**

**Reviewer #1:**
This study of mixed glutamate/GABA transmission from axons of the supramammillary nucleus to dentate gyrus seeks to sort out whether the two transmitters are released from the same or different synaptic vesicles. This conundrum has been examined in other dual-transmission cases and even in this particular pathway, there are different views. The authors use a variety of electrophysiological and immunohistochemical methods to reach the surprising (to me) conclusion that glutamate and GABA- filled vesicles are distinct yet released from the same nerve terminals. The strength of the conclusion rests on the abundance of data (approaches) rather than the decisiveness of any one approach, and I came away believing that the boutons may indeed produce and release distinct types of vesicles, but have reservations.

We thank the reviewer for his/her evaluation of our work. At present, several studies reported that a variety of combinations of two transmitters are co-released from different synaptic vesicles in the central nervous system. In this regard, we think the cotransmission of glutamate/GABA from different synaptic vesicles is not surprising. To better explain to the reader how much we know about co-release of dual transmitters in the brain, we have now added new sentences describing segregated co-release of two neurotransmitters in other synapses in the Introduction (line 63-80).

Accepting the conclusion, one is now left with another conundrum, not addressed even in the discussion: how can a single bouton sort out VGLUTs and VIAATs to different vesicles, position them in distinct locations with nm precision, and recycle them without mixing? And why do it this way instead of with single vesicles having mixed chemical content? For example, could a quantitative argument be made that separate vesicles allow for higher transmitter concentrations? I feel the paper needs to address these problems with some coherent discussion, at minimum.

Although these questions are very important and interesting to address, little is known about molecular mechanisms how VGluT2 and VIAAT are sorted to different vesicles and each synaptic vesicle is segregated. That is why we had not mentioned the sorting mechanisms in the original manuscript. Nevertheless, in response to the reviewer’s suggestion, we have now added new sentences describing possible mechanisms for the sorting and segregation of VGluT2 and VIAAT in the Discussion (line 439-462).

As for the question regarding why glutamate and GABA are released from different synaptic vesicles, we mentioned the functional roles of separate release of two transmitters over release from single vesicles several times in the Introduction (line 94100), Results (line 300-302), and Discussion (line 406-408, 521-522). Although it seems to be an interesting point to think about transmitter concentrations in the vesicles, we think this issue is beyond the scope of the present study. Given that manipulation of vesicular transmitter contents is technically possible (Hori and Takamori, 2021), this issue awaits further investigation.

Major concerns:(1) Throughout the paper, the authors use repetitive optogenetic stimulation to activate SuM fibers and co-release glutamate and GABA. There are several issues here: first, can the authors definitively assure the reader that all the short-term plasticity is presynaptic and not due to ChR2 desensitization? This has not been addressed. Second, can the authors also say that all the activated fibers release both transmitters? If for example 20% of the fibers retained a onetransmitter identity and had distinct physiological properties, could that account for some of the physiological findings?

Thank you for raising this important point. To examine whether repetitive light illumination induces ChR2 desensitization, the fiber volley was extracellularly recorded. We found that paired-pulse or 10 stimuli at 5, 10, and 20 Hz reliably evoked similar amplitudes of fiber volley during light stimulation. These results clearly indicate that repetitive light stimulation can reliably activate ChR2 and elicit action potentials in the SuM axons. These new findings are now included in Figure 1-figure supplement 2 and Figure 5-figure supplement 2. We also previously demonstrated that by direct patch-clamp recordings from ChR2-expressing hippocampal mossy fiber terminals, 125 times light stimulation at 25 Hz reliably elicited action potentials (Fig. S1: Fukaya et al., 2023). Therefore, we believe that if expression level of ChR2 is high, activation of ChR2 induces action potentials in response to repetitive light stimulation and mediates synaptic transmission with high efficiency.

We found that most of the SuM terminals (95%) have both VGluT2 and VIAAT (Figure 1E). This anatomical evidence strongly indicates that most of the SuM terminals have the ability to release both glutamate and GABA, and the SuM fibers having one transmitter identity should be minor populations.

(2) PPR differences in Figures 1F-I are statistically significant but still quite small. You could say they are more similar than different in fact, and residual differences are accounted for by secondary factors like differential receptor saturation.

In this experiment, the light intensity was adjusted to yield less than 80% of the maximum response as described in the method section of original and revised manuscript, minimizing the possibility of receptor saturation. We also excluded the possibility that PPR differences could be attributed to differential receptor saturation and desensitization by using a low-affinity AMPA receptor antagonist and a low-affinity GABAA receptor antagonist (Figure 5-figure supplement 3). These results indicate that PPR differences are mediated by the presynaptic origin.

(3) The logic of the GPCR experiments needs a better setup. I could imagine different fibers released different transmitters and had different numbers of mGluRs, so that one would get different modulations. On the assumption that all the release is from a single population of boutons, then either the mGluRs are differentially segregated within the bouton, or the vesicles have differential responsiveness to the same modulatory signal (presumably a reduced Ca current). This is not developed in the paper.

Based on our minimal stimulation results and anatomical analysis, we believe that many SuM terminals contain both glutamate and GABA. Therefore, both transmissions are able to be modulated by mGluRs and GABAB receptors within the same terminals. As the reviewer pointed out, differential responsiveness of glutamate-containing and GABA-containing vesicles to the GPCR signal could be one of the molecular mechanisms for differential effects of GPCRs on EPSCs and IPSCs. In addition, the spatial coupling between GPCRs and active zones for glutamate and GABA in the same SuM terminals may be different, which may give rise to differential modulation of glutamate and GABA release. These possible mechanisms are now described in the Discussion (line 469-476).

(4) The biphasic events of Figures 3 and S3: I find these (unaveraged) events a bit ambiguous. Another way to look at them is that they are not biphasic per se but rather are not categorizable. Moreover, these events are really tiny, perhaps generated by only a few receptors whose open probability is variable, thus introducing noise into the small currents.

We agree with the reviewer that some events are tiny and some small currents could be masked by background noise. We understand that detecting the biphasic events by minimal stimulation has technical limitations. Because we automatically detected biphasic events, which were defined as an EPSC-IPSC sequence, only if an outward peak current following an inward current appeared within 20 ms of light illumination as described in the method section, we cannot exclude the possibility that the biphasic events we detected might include false biphasic responses. To compensate these technical issues, we also performed strontium-induced asynchronous release as another approach and found similar results as minimal stimulation experiments (Figures 3E and 3F). Furthermore, we confirmed that the amplitudes and kinetics of minimal light stimulation-evoked EPSCs or IPSCs were not altered by blockade of their counterpart currents (Figure 3-figure supplement 2). Even if false biphasic responses were accidentally included in the analysis, eventually biphasic events are a minor population and we successfully detected discernible independent EPSCs and IPSCs, which were the major population of uniquantal release-mediated synaptic responses. Thus, multiple pieces of evidence support distinct release of glutamate and GABA from SuM terminals.

(5) Figure 4 indicates that the immunohistochemical analysis is done on SuM terminals, but I do not see how the authors know that these terminals come from SuM vs other inputs that converge in DG.

We thank the reviewer for raising an important point. As shown in Figure 4A, B, almost all VGluT2-positive terminals in the GC layer co-expressed with VIAAT. We are aware that VTA neurons reportedly project to the GC layer of the DG and co-release glutamate and GABA (Ntamati and Luscher, 2016). Contrary to this report, our retrograde tracing analysis did not reveal direct projections from the VTA to the DG. This new data is now included in Figure 4-figure supplement 1. We also added pre-embedding immunogold EM analysis, in which SuM terminals were virally labeled with eYFP, confirming that they form both asymmetric and symmetric synapses (revised Figure 4F). Together with these new data, our results clearly demonstrate that SuM terminals in the GC layer form both asymmetric and symmetric synapses. While our results strongly suggest that VGluT2positive terminals and SuM terminals in the GC layer are nearly identical, we cannot fully exclude the possibility that other inputs originating from unidentified brain regions may co-express VGluT2 and VIAAT in the GC layer. Therefore, in Figure 4 of the revised manuscript, we described “VGluT2-positive terminals” instead of “SuM terminals”.

(6) Figure 4E also shows many GluN1 terminals not associated with anything, not even Vglut, and the apparent numbers do not mesh with the statistics. Why?

In triple immunofluorescence for VGluT2, VIAAT, and GluN1, free GluN1 puncta were predominantly observed in the molecular layer. Given that VGluT2-positive terminals are sparse in the molecular layer, these GluN1 puncta are primarily associated with VGluT1, the dominant subtype. In this study, we focused the analysis of GluN1 puncta specifically on the GC layer, excluding the molecular layer. To avoid miscommunication, we changed the original Figure 4E to the new Figure 4G, which focuses on the GC layer and aligns with the quantitative analysis. Additionally, we used ultrathin sections (100-nm-thick) to enhance spatial resolution, which limits the detection of co-localization events within this confined spatial range, as noted in the Discussion (line 485-488).

(7) Do the conclusions based on the fluorescence immuno mesh with the apparent dimensions of the EM active zones and the apparent intermixing of labeled vesicles in immuno EM?

To further support our immunofluorescence results, we performed EM study and found that a single SuM terminal formed both asymmetric and symmetric synapses on a GC soma (revised Figures 4E and 4F). These new data and our immunofluorescence results clearly indicate that a single SuM terminal forms both glutamatergic and GABAergic synapses on a GC and co-release glutamate and GABA.

As the reviewer pointed out, our immuno EM shows that VGluT2 and VIAAT labeled vesicles appear to intermix in asymmetric and symmetric synapses. Accordingly, in the revised manuscript, Figure 7 has been modified to show the intermixing of glutamate and GABA-containing vesicles in the SuM terminal. It should be noted that because of low labeling efficiency, our immuno-EM images don’t represent the whole picture of synaptic vesicles for glutamate and GABA. There could be biased distribution of vesicles close to their release site (more VGluT2-containing vesicles close to asymmetric synapses and more VIAAT-containing vesicles close to symmetric synapses) as reported previously (Root et al., 2018). Additionally, our results could be explained by other mechanisms: co-release of glutamate and GABA from the same vesicles, with one transmitter undetected due to the absence of its postsynaptic receptor. This possibility is now mentioned in the Discussion (line 512-520). More detailed vesicle configuration in a single SuM terminal will have to be investigated in future studies.

(8) Figure 6 is not so interesting to me and could be removed. It seems to test the obvious: EPSPs promote firing and IPSPs oppose it.

We believe these results are necessary for the following two reasons. First, we showed that glutamate/GABA co-transmission balance is dynamically changed in a frequency-dependent manner (Figure 5). In terms of physiological significance, it is important to demonstrate how these frequency-dependent dynamic changes affect GC firing. Therefore, we believe that figure 6, which shows how SuM inputs modulate GC firing by repetitive SuM stimulation, is necessary for this paper. Second, we previously reported the excitatory effects of the SuM inputs on GC firing, suggesting the important roles of glutamatergic transmission of the SuM inputs in synaptic plasticity (Hashimotodani et al., 2018; Hirai et al., 2022; Tabuchi et al., 2022). In contrast, how GABAergic cotransmission contributes to SuM-GC synaptic plasticity and DG information processing was not well understood. Our results in figure 6, which demonstrate the inhibitory effects of GABAergic co-transmission on GC firing by high frequency repetitive SuM input activity, clearly show the contribution of GABAergic co-transmission to short-term plasticity at SuM-GC synapses. For these reasons, we would like to keep Figure 6. We hope that our explanations convince the reviewer.

**Reviewer #2:**
Summary:In this study, the authors investigated the release properties of glutamate/GABA co-transmission at the supramammillary nucleus (SuM)-granule cell (GC) synapses using in vitro electrophysiology and anatomical approaches at the light and electron microscopy level. They found that SuM to dentate granule cell synapses, which co-release glutamate and GABA, exhibit distinct differences in paired-pulse ratio, Ca2+ sensitivity, presynaptic receptor modulation, and Ca2+ channel-vesicle coupling configuration for each neurotransmitter. The study shows that glutamate/GABA co-release produces independent glutamatergic and GABAergic synaptic responses, with postsynaptic targets segregated. They show that most SuM boutons form distinct glutamatergic and GABAergic synapses in close proximity, characterized by GluN1 and GABAAα1 receptor labeling, respectively. Furthermore, they demonstrate that glutamate/GABA co-transmission exhibits distinct short-term plasticity, with glutamate showing frequencydependent depression and GABA showing frequency-independent stable depression.Their findings suggest that these distinct modes of glutamate/GABA co-release by SuM terminals serve as frequency-dependent filters of SuM inputs.Strengths:The conclusions of this paper are mostly well supported by the data.

We thank the reviewer for their positive and constructive comments on our manuscript.

Weaknesses:Some aspects of Supplementary Figure 1A and the table need clarification. Specifically, the claim that the authors have stimulated an axon fiber rather than axon terminals is not convincingly supported by the diagram of the experimental setup. Additionally, the antibody listed in the primary antibodies section recognizes the gamma2 subunit of the GABAA receptor, not the alpha1 subunit mentioned in the results and Figure 4.

We have now answered these questions in recommendations section below.

**Reviewer #3:**
Summary:In this manuscript, Hirai et al investigated the release properties of glutamate/GABA cotransmission at SuM-GC synapses and reported that glutamate/GABA co-transmission exhibits distinct short-term plasticity with segregated postsynaptic targets. Using optogenetics, whole-cell patch-clamp recordings, and immunohistochemistry, the authors reveal distinct transmission modes of glutamate/GABA co-release as frequency-dependent filters of incoming SuM inputs.Strengths:Overall, this study is well-designed and executed; conclusions are supported by the results. This study addressed a long-standing question of whether GABA and glutamate are packaged in the same vesicles and co-released in response to the same stimuli in the SuM-GC synapses (Pedersen et al., 2017; Hashimotodani et al., 2018; Billwiller et al., 2020; Chen et al., 2020; Li et al., 2020; Ajibola et al., 2021). Knowledge gained from this study advances our understanding of neurotransmitter co-release mechanisms and their functional roles in the hippocampal circuits.Weaknesses:No major issues are noted. Some minor issues related to data presentation and experimental details are listed below.

We appreciate the reviewer’s positive view of our study. We responded in more detail in recommendations section below.

**Recommendations for the authors:**

**Reviewer #1:**
(1) The blue color for VIAAT in panel 1C is extremely hard to see.

Thank you for pointing out. We have changed to the cyan color for VIAAT in Figure 1C and D in the revised manuscript.

(2) Line 329 "perforant" not "perfomant".

We appreciate the reviewer’s careful attention. In the revised manuscript, we corrected this misword.

**Reviewer #2:**
To convincingly demonstrate that the authors stimulated SuM axon fiber instead of SuM terminals (Supplementary Figures 1A), they should provide an image showing the distribution of SuMlabeled fibers and axon terminals reaching the dentate gyrus (DG) and the trace of the optic fiber, rather than providing a diagram of the experimental setup.

We appreciate the reviewer’s suggestion. We have now provided a new experimental setup image (Figure 1-figure supplement 1A) showing a single GC, the distribution of SuM fibers in the GC layer, and the illumination area at each location. As SuM inputs make synapses onto the GC soma and dendrite close to the GC cell body, SuM-GC synapses in the recording GCs exist in a very limited area. This characteristic synaptic localization allowed us to control the illumination area without applying light to the SuM terminals in the recording GCs. Delayed onsets of EPSCs/IPSCs by over-axon stimulation (Figure 1-figure supplement 1C, D) also support that SuM terminals in the recording GCs were out of illumination area.

Additionally, the authors should clarify the discrepancy between the antibody mentioned in the list of primary antibodies, which recognizes the gamma2 subunit of the GABAA receptor, and the alpha1 subunit of the GABAA receptor mentioned in the results and Figure 4.

We apologize for this mistake. As described in the main text and figure, we used the antibody for a1 subunit of the GABAA receptor. Table S1 has been corrected in the revised version of the paper.

**Reviewer #3:**
(1) In Figure 1, the authors used two [Ca2+]o concentrations to study the EPSC and IPSC amplitudes. How does the Ca2+ concentration affect the PPR in the EPSC and IPSC, respectively?

Given that lowering the extracellular Ca2+ concentration reduces the release probability, it is expected that 1 mM extracellular Ca2+ concentration increases PPR compared to 2.5 mM. Actually, we observed that lowering the extracellular Ca2+ concentration increased the synaptic responses from 2nd to 10th (both EPSC and IPSC) by train stimulation (Figure 5).

(2) In Figure 2D, does baclofen also have a dose-dependent effect on the inhibition of the EPSC and IPSC similar to the DCG-IV in Figure 2C?

Thank you for your question. Because we aimed to demonstrate the differential inhibitory effects of baclofen at a certain concentration on glutamatergic and GABAergic co-transmission, we did not go into detail regarding a dose-dependent effect. In response to the reviewer’s comment, we performed the effects of higher concentration of baclofen on EPSCs and IPSCs. As shown in the figure below, 50 µM baclofen inhibited EPSCs and IPSCs to the similar extent. Therefore, by comparing inhibitory effect of two different concentrations of baclofen (5 and 50 µM), we believe that baclofen also has a dose-dependent inhibitory effect on both EPSCs and IPSCs similar to the DCGIV.

**Author response image 1. sa4fig1:** 

(3) In Figure 2E, statistical labels, such as "*" or "n.s." (not significant), should be provided on the plots to facilitate the reading of figures.

In response to the reviewer’s comment, we have provided statistical labels in the Figure 2E.

(4) In Figure 3A, the latency of the evoked EPSC for the lower light stimulation groups seems to be much slower than the one shown on the left or other figures in the paper, such as Figure 1F.

Please double-check if the blue light stimulation label is placed in the right location.

Corrected, thanks.

(5) The use of minimal light stimulation in optogenetic experiments is not appropriately justified or described. More detailed information should be provided, such as whether the optogenetic stimulation is performed on the axon or the terminals of the SuM.

We appreciate the reviewer’s suggestion. To effectively detect stochastic synaptic responses, the light stimulation was applied on the terminals of the SuM. We have now stated this information (line 212). We also further described the justification of use of minimal light stimulation in the revised manuscript (line 207-209).

References

Fukaya R, Hirai H, Sakamoto H, Hashimotodani Y, Hirose K, Sakaba T (2023) Increased vesicle fusion competence underlies long-term potentiation at hippocampal mossy fiber synapses. Sci Adv 9:eadd3616.

Hashimotodani Y, Karube F, Yanagawa Y, Fujiyama F, Kano M (2018) Supramammillary Nucleus Afferents to the Dentate Gyrus Co-release Glutamate and GABA and Potentiate Granule Cell Output. Cell Rep 25:2704-2715 e2704.

Hirai H, Sakaba T, Hashimotodani Y (2022) Subcortical glutamatergic inputs exhibit a Hebbian form of long-term potentiation in the dentate gyrus. Cell Rep 41:111871.

Hori T, Takamori S (2021) Physiological Perspectives on Molecular Mechanisms and Regulation of Vesicular Glutamate Transport: Lessons From Calyx of Held Synapses. Front Cell Neurosci 15:811892.

Ntamati NR, Luscher C (2016) VTA Projection Neurons Releasing GABA and Glutamate in the Dentate Gyrus. eNeuro 3.

Root DH, Zhang S, Barker DJ, Miranda-Barrientos J, Liu B, Wang HL, Morales M (2018) Selective Brain Distribution and Distinctive Synaptic Architecture of Dual Glutamatergic-GABAergic Neurons. Cell Rep 23:3465-3479.

Tabuchi E, Sakaba T, Hashimotodani Y (2022) Excitatory selective LTP of supra-mammillary glutamatergic/GABAergic co-transmission potentiates dentate granule cell firing. Proc Natl Acad Sci U S A 119:e2119636119.